# Single-cell RNA-seq reveals trans-sialidase-like superfamily gene expression heterogeneity in *Trypanosoma cruzi* populations

**Lucas Inchausti[1,2], Lucia Bilbao[1,2], Vanina A Campo[3,4], Joaquín Garat[5], José Sotelo-Silveira[5,6], Gabriel Rinaldi[7], Virginia M Howick[8], Maria A Duhagon[2], Javier G De Gaudenzi[3,4], Pablo Smircich[1,2]***

[1]Laboratorio de Bioinformática, Departamento de Genómica, Instituto de Investigaciones Biológicas Clemente Estable, Montevideo, Uruguay; [2]Sección Genómica Funcional, Facultad de Ciencias, Universidad de la República, Montevideo, Uruguay; [3]Instituto de Investigaciones Biotecnológicas, Universidad Nacional de San Martín-Consejo Nacional de Investigaciones Científicas y Técnicas, Buenos Aires, Argentina; [4]Escuela de Bio y Nanotecnologías (EByN), Universidad Nacional de San Martín, Buenos Aires, Argentina; [5]Departamento de Genómica, Instituto de Investigaciones Biológicas Clemente Estable, Montevideo, Uruguay; [6]Sección Biología Celular, Facultad de Ciencias, Universidad de la República, Montevideo, Uruguay; [7]Department of Biology, University of Oxford, Oxford, United Kingdom; [8]School of Biodiversity, One Health and Veterinary Medicine, University of Glasgow, Glasgow, United Kingdom

**\*For correspondence:**
psmircich@fcien.edu.uy

## eLife Assessment

This study provides **important** insights into how *Trypanosoma cruzi* populations diversify surface protein expression, showing through single-cell RNA sequencing that trans-sialidase-like genes are expressed heterogeneously across individual parasites, a pattern with clear implications for immune evasion. The evidence is **convincing**, supported by robust single-cell transcriptomic analyses, consistent quantitative measures of expression heterogeneity, and integration with genomic organization that together argue against purely stochastic expression.

**Abstract** *Trypanosoma cruzi,* the causative agent of Chagas disease, presents a major public health challenge in Central and South America, affecting approximately 8 million people and placing millions more at risk. The *T. cruzi* life cycle includes transitions between epimastigote, metacyclic trypomastigote, amastigote, and blood trypomastigote stages, each marked by distinct morphological and molecular adaptations to different hosts and environments. Unlike other trypanosomatids such as *Trypanosoma brucei*, *T. cruzi* does not employ a monoallelic model of antigenic variation; instead, it relies on a diverse repertoire of cell-surface associated proteins encoded by large multigene families, which are essential for infectivity and immune evasion. This study analyzes cell-specific transcriptomes using single-cell RNA sequencing of amastigote and trypomastigote cells to characterize stage-specific surface protein expression during mammalian infection. Through clustering and identification of cell-specific markers, we assigned cells to distinct parasite developmental forms. Analysis of individual cells revealed that surface protein-coding genes, especially members of the trans-sialidase-like superfamily (TcS), are expressed with greater heterogeneity than single-copy

genes. Moreover, no recurrent combinations of TcS genes were observed between individual cells in the population. Remarkably, a small subset of TcS mRNAs, encoded by genes preferentially located in the core genomic compartment, are frequently detected across the cell population, whereas the vast majority of TcS mRNAs show low detection frequencies and are mainly encoded in the disruptive compartment. Our findings thus reveal transcriptomic heterogeneity within trypomastigote populations where each cell displays unique TcS expression profiles. Focusing on the diversity of surface protein expression, this research aims to deepen our understanding of *T. cruzi* cellular biology and infection strategies.

## Introduction

The protozoan parasite *Trypanosoma cruzi* is the etiological agent of Chagas disease, a highly prevalent infectious disease in Central and South America that affects approximately 8 million people, with several million more at risk of infection (*Tarleton, 2016*).

The parasite life cycle involves both an invertebrate vector (triatomine bug) and a vertebrate host. The change in environmental conditions triggers differentiation processes in the parasite developing across four main stages. The epimastigote form replicates within the insect and differentiates into metacyclic trypomastigotes in the insect's rectal tract. This latter form does not replicate, specializes in host infection, and can invade various cell types. Once inside the cell, the parasite differentiates into an amastigote, the replicative form within the mammalian host. After several rounds of division, amastigotes differentiate into bloodstream trypomastigotes, which, after cell lysis, can either infect new cells or be ingested by the vector (*Martín-Escolano et al., 2022*). Parasite developmental stages exhibit significant morphological and molecular differences, associated with distinct gene expression profiles. In trypanosomatids, protein-coding genes are organized into large polycistronic RNA polymerase II transcription units, with limited evidence for tight, gene-specific transcriptional control (*Clayton, 2019*). As a result, gene expression is largely regulated at the post-transcriptional level, through mechanisms controlling mRNA steady-state abundance, translation, and related processes (*Clayton, 2019*). More recently, accumulating evidence indicates that genome organization and chromatin state also shape transcriptional outputs (*Díaz-Viraqué et al., 2023*; *Lima et al., 2021*; *Lima et al., 2022*; *Beati et al., 2023*; *Ocampo et al., 2025*). These studies have identified two distinct chromatin compartments in *T. cruzi*: a core compartment, in which genes tend to show more uniform expression consistent with permissive transcription and predominant post-transcriptional regulation (*Berná et al., 2018*), and a disruptive compartment, enriched in multigene families, whose genes exhibit lower average transcription and stronger chromatin-associated regulation (*Díaz-Viraqué et al., 2023*).

*T. cruzi* infection relies on a heterogeneous set of membrane proteins, encoded mainly by large multigene families (*Belew et al., 2017*). Among these are trans-sialidases and trans-sialidase-like proteins (TcS), mucins, MASPs, GP63, DGFs, and RHS proteins, most of which are involved in infection, tropism, and immune evasion (*de la Pech-Canul et al., 2017*; *Freitas et al., 2011*; *Buscaglia et al., 2006*; *Pereira-Chioccola et al., 2000*; *de Pablos et al., 2011*; *Bartholomeu et al., 2009*).

The trans-sialidase-like superfamily is involved in processes underlying host-parasite interactions (*Freitas et al., 2011*). Members that retain enzymatic activity catalyze the transfer of sialyl groups from host glycoconjugates to galactopyranosyl units on the parasite's surface, an essential activity, as *T. cruzi* is unable to synthesize de novo sialic acid (*de Burle-Caldas et al., 2022*). This is the largest superfamily, with over 1,400 members, and is subdivided into eight groups based on amino acid sequence (*Freitas et al., 2011*). Few members have been functionally characterized, most expressed primarily in mammalian stages (*Berná et al., 2017*). The TcS group I includes proteins that retain trans-sialidase activity, though members from all groups are involved in host-parasite interactions (*Tonelli et al., 2010*).

In recent years, single-cell RNA sequencing (scRNA-seq) has been employed in protozoa, with reports including *Plasmodium falciparum*, *Trypanosoma brucei,* and *Leishmania major* (*Walzer et al., 2018*; *Ngara et al., 2018*; *Reid et al., 2018*; *Howick et al., 2019*; *Vigneron et al., 2020*; *Hutchinson et al., 2021*; *Briggs et al., 2021*; *Howick et al., 2022*; *Quintana et al., 2022*; *Catta-Preta et al., 2024*). These studies revealed key aspects of the infection process undetectable with conventional methods, highlighting the relevance of this approach for understanding individual variation in gene

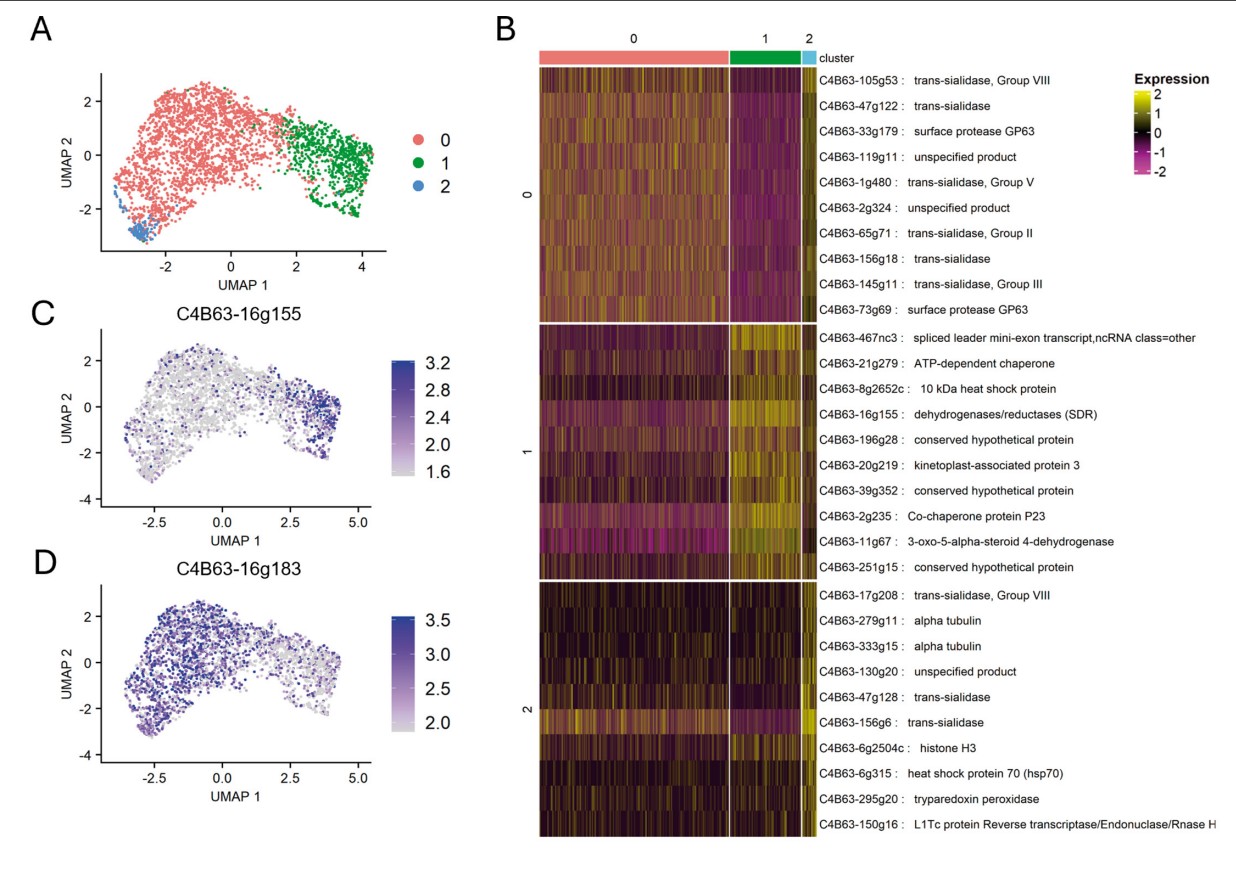

**Figure 1.** Identification of amastigote and trypomastigote cell populations. (**a**) UMAP colored by detected clusters based on gene expression profiles. (**b**) Heatmap of the top 10 gene markers upregulated in each of the three-cell populations identified (log2FC >1 and adjusted p-value <0.05). (**c**) Expression of a cluster 0 marker gene (C4B63-16g183) on the UMAP, and (**d**) Expression of a cluster 1 marker gene (C4B63-16g155) on the UMAP.

expression in single-celled organisms (*Abuchery et al., 2022*). In *T. brucei,* antigenic variation driven by variant surface glycoproteins (VSGs) has been studied at single-cell resolution to understand the mechanisms that enable subpopulations of this parasite to evade the immune system, as a bet hedging strategy, ensuring parasite survival (*Hutchinson et al., 2021*). In this parasite, scRNA-seq revealed that pre-metacyclic cells express multiple VSG transcripts simultaneously, in contrast to metacyclic forms, which display a protein coat composed of a single VSG type.

While population-level heterogeneity in surface protein expression has been suggested as critical for *T. cruzi* infection, immune evasion, and persistence (*Seco-Hidalgo et al., 2015*; *Luzak et al., 2021*), this has not been studied at the intra-population level, and the underlying mechanisms remain poorly understood. Here, we present a scRNA-seq analysis of *T. cruzi* to understand the heterogeneity in surface protein expression within trypomastigote populations.

## Results and discussion
### Identification of cell populations

To assess the expression of cell-surface protein-coding genes in *T. cruzi*, we conducted a 10X Chromium Single Cell 3' assay from a mixed population of amastigotes and cell-derived trypomastigotes, aiming at sequencing the transcriptome of 5000 cells. After low-quality cell filtering and gene expression quantification (see Materials and methods), we obtained 3192 single-cell transcriptomes with 14321 total genes detected, with a mean of 1088 genes and 2461 UMIs detected per cell. These results were comparable to other scRNA-seq studies done in *T. brucei* and more recently addressed in *T. cruzi* (currently reported as a preprint) using 10X Chromium technology (*Briggs et al., 2021*; *Li et al., 2016*; *Laidlaw et al., 2025*). Cell populations (*Figure 1a*) were defined by identifying cluster-specific

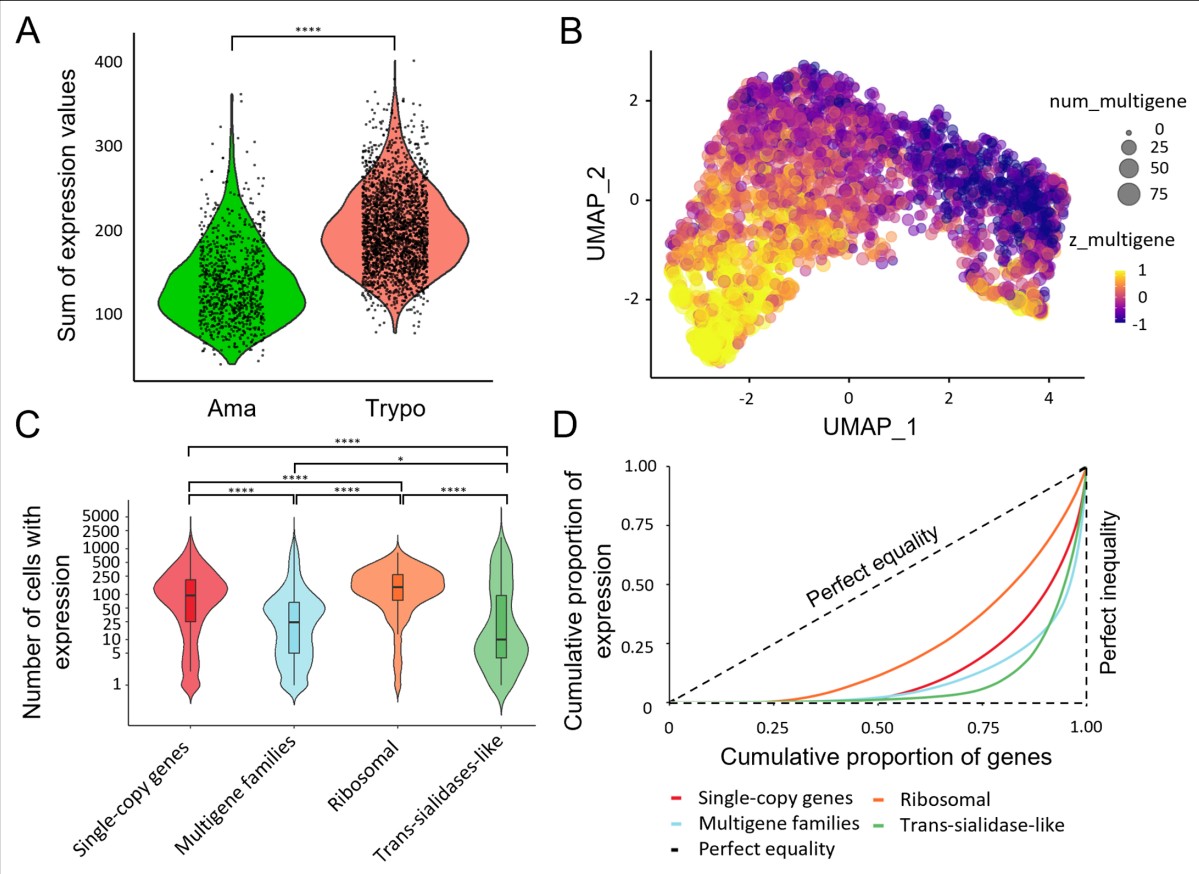

**Figure 2.** Overview of expression patterns across amastigote and trypomastigote cells. (**a**) Summation of expression levels values from all multigene family genes for each cell from amastigote (Cluster 1) and trypomastigote (Cluster 0) cell populations (**** $p<0.0001$, mean$_{Ama}$ = 137.2, SD$_{Ama}$ = 48.7, mean$_{Trypo}$ = 201.1, SD$_{Trypo}$ = 48.6, FC$_{Trypo/Ama}$ = 1.5). (**b**) UMAP visualization of the expression patterns of multigene family genes; num_multigene indicates the number of multigene family genes detected per cell (genes with >0 UMI counts). Z_multigene reflects the relative expression level of multigene family genes per cell, calculated as the z-score-standardized sum of their UMI counts, such that positive values reflect above-average multigene family expression and negative values reflect below-average levels. (**c**) Violin plots showing the number of cells expressing a specific gene belonging to each group of genes: subsampled single-copy and multigene families, ribosomal genes, and trans-sialidases. To avoid biases against size differences between single-copy and multigene family genes, we generated a subsampled single-copy genes list, randomly selecting an equal number of genes as those from the multigene family's gene set. The expression distribution of the subsampled single-copy genes is similar to the distribution of the entire dataset (* $p<0.05$, **** $p<0.0001$. See ***Supplementary file 3***). (**d**) Lorenz curves showing the cumulative proportion of gene expression relative to the cumulative proportion of genes for subsampled single-copy, multigene family genes, ribosomal protein coding genes, and trans-sialidase-like genes. Genes were ordered by total expression, and the dashed line indicates perfect equality. Curves that deviate further from the diagonal reflect greater inequality, meaning that fewer genes account for most of the expression within each category. Statistically significant differences between groups for **c** and **d** are shown in ***Supplementary file 3***.

The online version of this article includes the following figure supplement(s) for figure 2:

**Figure supplement 1.** Extended overview of expression patterns across amastigote and trypomastigote cells.

gene markers (***Figure 1b*** and ***Supplementary file 1***). Two cell clusters were assigned to trypomastigotes and amastigotes: cluster 0 (2201 cells) and cluster 1 (824 cells), respectively. Markers gene expression was consistent with previous bulk RNA-seq data from Dm28c (***Díaz-Viraqué et al., 2023***; ***Figure 1b, c and d***, ***Supplementary file 2***). In turn, we hypothesize that cluster 2, which comprised only 167 cells, reflects amastigote-trypomastigote transitioning parasites, as its gene markers are differentially expressed in bulk RNA-seq data, but some are upregulated in amastigotes and others in trypomastigotes (***Supplementary file 2***).

## Expression pattern of surface protein-coding genes

We analyzed differences in gene expression between single-copy genes and multigene families, between and within the identified cell populations. As previously reported (***Díaz-Viraqué et al., 2023***;

*Berná et al., 2017*), multi-gene family expression is increased in trypomastigotes (*Figure 2a*), consistent with the involvement of these genes in stage-specific functions. As a control, we examined the expression of single-copy genes, for which this pattern was not observed as these genes are primarily associated with basic cellular functions and show more similar average expression levels between the two cell types (*Figure 2—figure supplement 1a*; *Li et al., 2016*).

Within trypomastigotes, heterogeneous expression of surface protein-coding gene families was observed, with high variation (compared to single-copy genes) in the number of cells in which each surface protein-coding gene was detected, as well as the total expression level in each cell (*Figure 2b and c*). Even though expression heterogeneity is also observed for single-copy genes (*Figure 2—figure supplement 1a* and *Figure 1b*), probably due to the sampling biases that cause gene dropout in 10 X Genomics technology, we investigated whether this phenomenon was more pronounced in surface multigene families. Therefore, we analyzed differences among single-copy and multigene family genes (together or grouped by multigene family) in trypomastigotes, in terms of the number of cells expressing each of the individual genes of each group (*Figure 2c*, *Figure 2—figure supplement 1c* and *Supplementary file 3*). In addition, we assessed expression inequality using Lorenz curves (*Figure 2d*, *Figure 2—figure supplement 1d* and *Supplementary file 3*) to evaluate how unevenly gene expression is distributed within each gene group. Ribosomal protein–coding genes were included as a control group.

Interestingly, compared to single-copy genes, especially ribosomal genes, multigene family genes showed a greater dispersion regarding the number of cells in which each gene was detected (*Figure 2c*, *Figure 2—figure supplement 1c* and *Supplementary file 3*), as well as a more pronounced deviation from the diagonal in the Lorenz curves (*Figure 2d*, *Figure 2—figure supplement 1d*), which represents perfect equality (reflected by different Gini indexes, see *Supplementary file 3*). Both observations indicate a higher expression heterogeneity for surface protein expression in the trypomastigote population. When multigene families were analyzed separately, several showed significant differences when compared to single-copy genes. Heterogeneity of the MASP family has already been suggested by studying clonal populations supporting our single-cell results (*Seco-Hidalgo et al., 2015*). Also, TcS genes exhibited a pronounced expression heterogeneity (*Figure 2—figure supplement 1c and d*). TcS constitutes a well-established and biologically central gene family in *T. cruzi*, playing key roles in host-parasite interactions (*de la Pech-Canul et al., 2017*; *Freitas et al., 2011*; *de Burle-Caldas et al., 2022*). In this context, we next focused on a more detailed characterization of TcS expression heterogeneity in trypomastigotes.

## Expression pattern of the TcS superfamily

When we re-clustered the trypomastigote population based solely on TcS gene expression, we identified two sub-populations: trypomastigote cluster 0 ('Trypo_0' composed of 1186 cells), which overexpressed these genes compared to trypomastigote cluster 1 ('Trypo_1' composed of 1015 cells; *Figure 3a*).

The two trypomastigote sub-populations segregated by TcS expression show only slight differences in the expression of other gene categories, such as ribosomal protein-coding genes (FC = 1.03), transporters (FC = 1.09), polymerase-related genes (FC = 1.15), and phosphatases (FC = 1.10) (*Figure 3b* and *Figure 3—figure supplement 1*, fold change values correspond to Trypo_0/Trypo_1 expression ratios). Although there is a tendency for surface protein genes to be more expressed in cluster Trypo_0 (FC = 1.38), trans-sialidases displayed the highest fold change between the two trypomastigote subpopulations (FC = 1.53; *Figure 3c* and *Figure 3—figure supplement 1*). It is tempting to speculate that this may reflect different infectivity amongst trypomastigote subpopulations, consistent with reports of 'broad' and 'slender' forms (*Schmatz et al., 1983*). Recently, while this manuscript was being prepared, a similar approach by Laidlaw et al. described the phenomenon by clustering trypomastigote cells based on the expression of all genes (*Laidlaw et al., 2025*). When applying this strategy to our data, their observation was reproduced.

When all cells are considered, most TcS genes are expressed, but in each individual cell, only approximately 40 TcS genes were detected. Interestingly, we observed that both trypomastigote subpopulations contain a subgroup of TcS genes that are detected in a large portion of cells (>40%; *Figure 4a*, and *Supplementary file 4*), indicating high-level expression at the population level. This subset comprises 31 TcS genes belonging to subfamilies II-VI and VIII (*Supplementary file 4*).

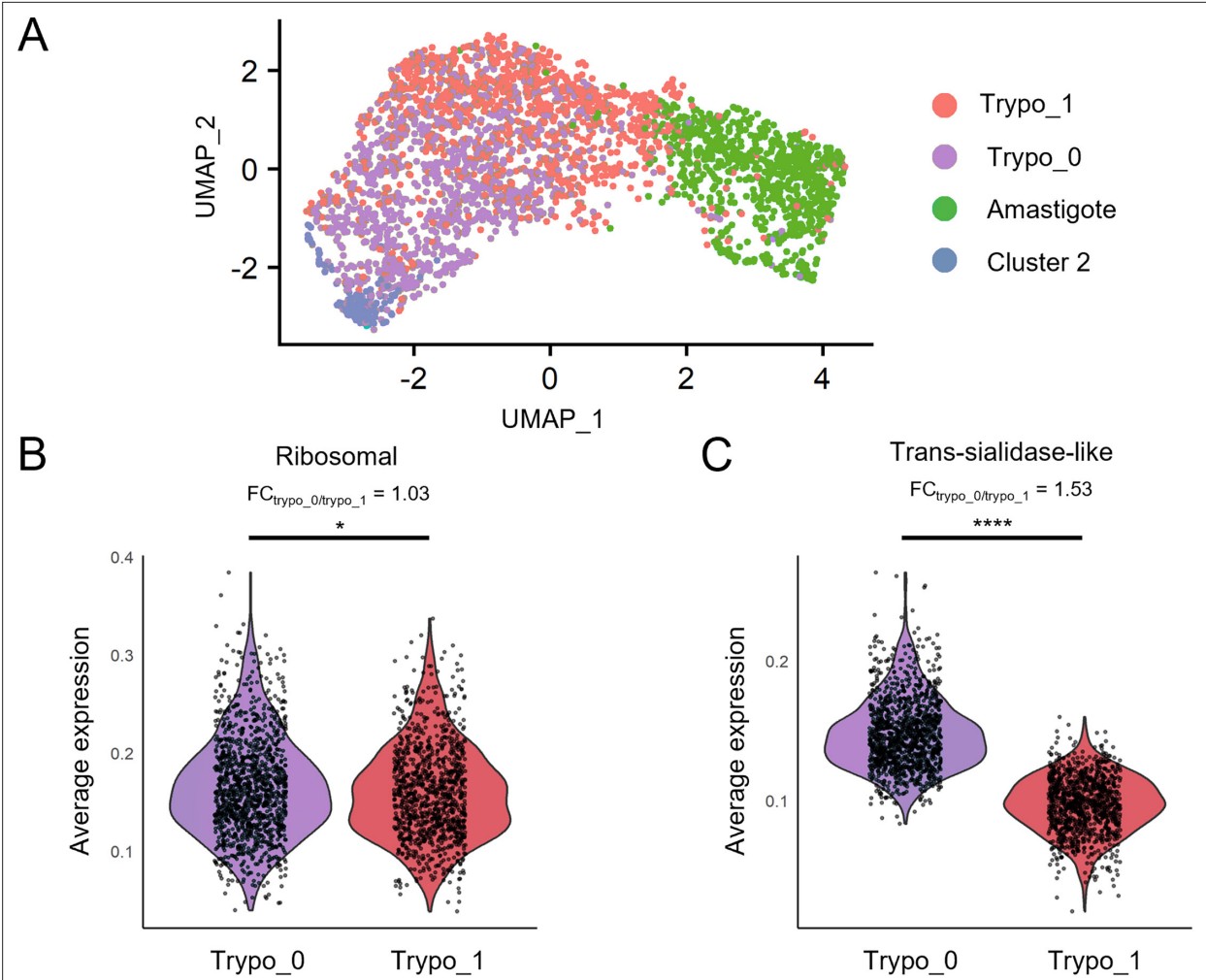

**Figure 3.** Trypomastigote subpopulations identified based on trans-sialidase expression profiles. (a) UMAP visualization colored by detected clusters based on gene expression profiles, with trypomastigote subpopulations identified. (b) violin plot displaying average expression levels of ribosomal protein-coding genes across sub-populations. (c) violin plot showing combined trans-sialidase expression levels for each sub-population. * p<0.05, **** p<0.0001.

The online version of this article includes the following figure supplement(s) for figure 3:

**Figure supplement 1.** Trypomastigote sub-clusters identified based on trans-sialidase expression profiles.

Consistent with our findings, *Laidlaw et al., 2025* reported a similar phenomenon, detecting on average approximately 40 TcS genes per cell, with a subset that is frequently detected across the trypomastigote population, further supporting the reproducibility of scRNA-seq results across studies. Gene dropouts in scRNA-seq experiments can generate apparently stochastic detection patterns for multigene family members, particularly affecting low-abundance transcripts, and thus represent an important potential confounder in the analysis of multigene families. To evaluate whether this technical effect could account for the observed TcS detection patterns, we examined the relationship between detection frequency and expression level. If random sampling were the primary driver, genes with similar average expression levels would be expected to exhibit comparable detection frequencies. However, we observed that many TcS genes with similar average expression are detected in markedly different proportions of cells (*Figure 4b*), arguing against a purely stochastic dropout model. Notably, among the 50 TcS genes with the highest average expression across cells, 62% are detected in fewer than 5% of cells (*Figure 4b*).

 At the single-cell level, we found that detected TcS genes contribute relatively evenly to the total family expression within each cell, as indicated by the low average Gini index of TcS expression levels

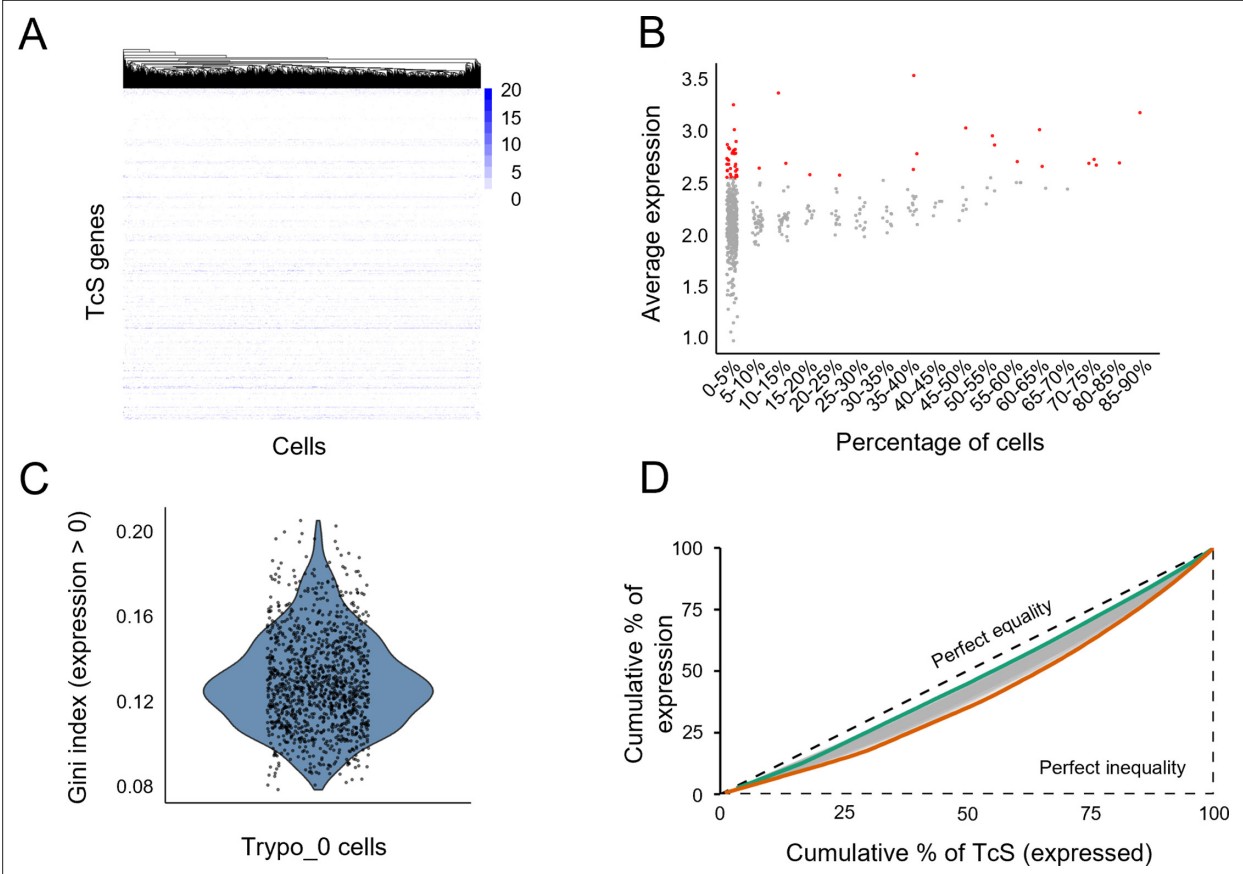

**Figure 4.** Overview of TcS gene expression patterns in Trypo_0 cells. (**a**) Heatmap displaying the expression of TcS genes in each cell that together account for 75% of total TcS gene expression within cluster Trypo_0. Cells are clustered by TcS expression profiles, with colors representing each gene's percentage contribution to the cell's total TcS expression. (**b**) Average expression of TcS genes grouped by the percentage of cells expressing each gene. In red are highlighted the top 50 TcS with highest average expression. (**c**) Gini index distribution for trypomastigotes cluster 0 (Trypo_0) cells considering only TcS detected in each cell[1] (**d**) Lorenz curves showing, for each cell in cluster Trypo_0, the cumulative proportion of total TcS expression as a function of the cumulative proportion of detected TcS genes. Genes were ordered by total expression, and the dashed line indicates perfect equality (i.e. all detected TcS genes contribute equally to the total TcS expression of a given cell). Green and orange curves correspond to cells with higher and lower expression equality, respectively.

The online version of this article includes the following figure supplement(s) for figure 4:

**Figure supplement 1.** Overview of gene expression of high frequency TcS.

across each cell (*Figure 4c and d*). This suggests that, within each cell, total TcS expression is shared across the detected TcS genes rather than being dominated by a few very highly expressed transcripts in that cell. Consistent with this interpretation, most TcS genes detected in a high fraction of cells in our scRNA-seq dataset also rank among the most highly expressed TcS genes in an independent bulk RNA-seq study (*Figure 4—figure supplement 1a*). This highlights a key limitation of bulk RNA-seq, as it may wrongly indicate that a set of few genes are highly expressed in each cell, while in fact, TcS expression is evenly distributed among all detected family members, regardless of the number of cells expressing each gene. Taken together with the results in *Figure 4b*, this observation further supports the conclusion that the observed expression pattern is unlikely to arise from a purely stochastic detection of TcS mRNAs.

While our data indicate transcriptional heterogeneity, it is important to note that future studies will be required to determine the extent to which this diversity translates to protein expression on the parasite surface. Consistent with our observations, recent proteomic analyses (*Cruz-Saavedra et al., 2025*) report pronounced variability in the expression of surface proteins, including TcS, suggesting that the heterogeneous transcriptional patterns observed here likely reflect biologically relevant differences.

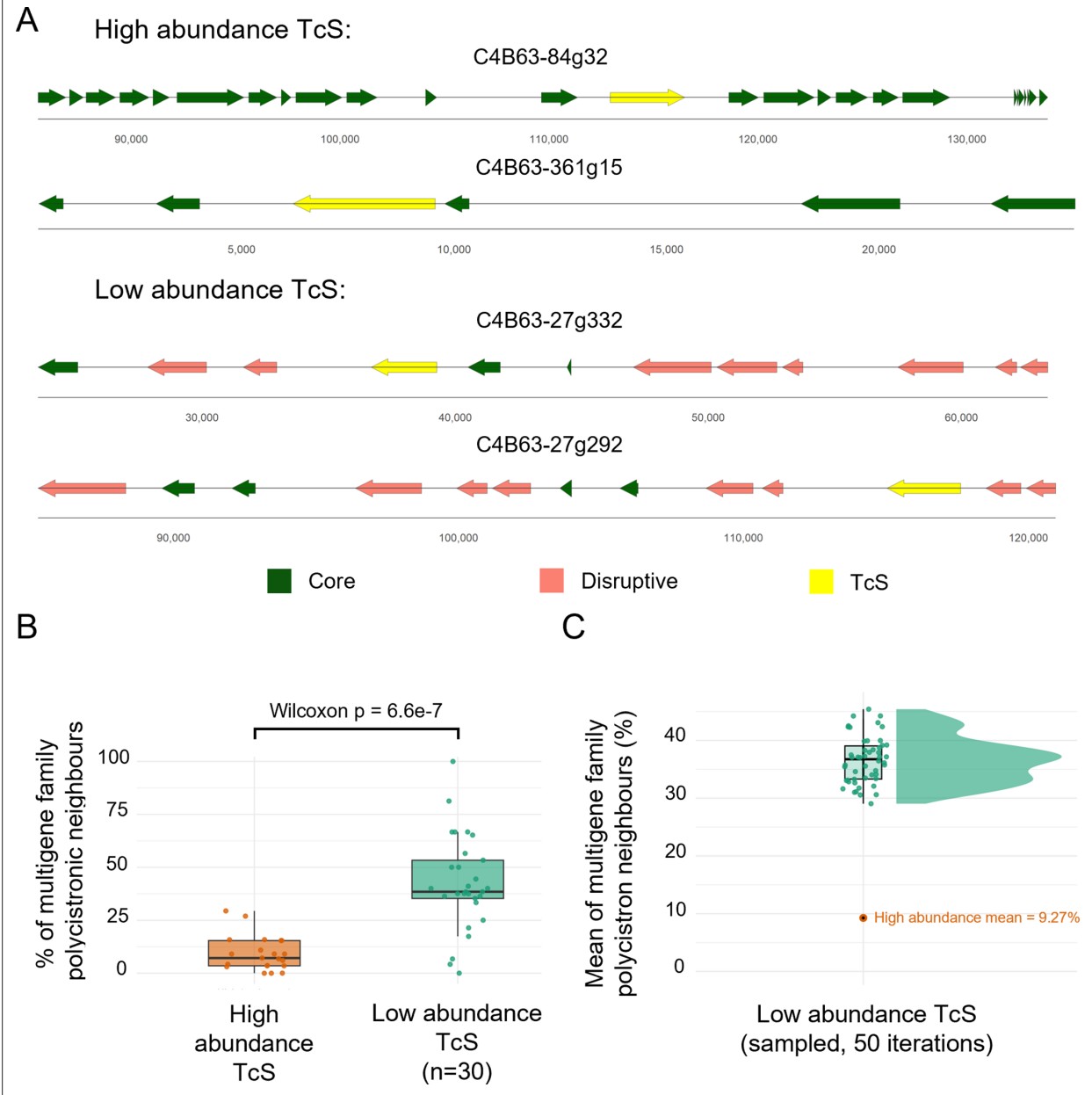

**Figure 5.** Genomic context and neighborhood composition of frequently detected versus lowly detected TcS loci. (**a**) Representative genomic loci of frequently detected (top, high abundance) and lowly detected (bottom, low abundance) TcS genes. Genes are shown as arrows, colored according to genomic compartment: core (dark green), disruptive (salmon), and TcS genes under analysis (yellow). Chromosomal coordinates are indicated below each locus. (**b**) Comparison of the percentage of multigene-family neighbors within polycistronic transcription units containing frequently detected and lowly detected TcS genes. Lowly detected TcS genes were subsampled to n=30. Wilcoxon rank-sum test: p=6.6 × 10⁻⁷. (**c**) Mean percentage of multigene-family neighbors in polycistrons calculated from 50 random subsets (n=30) of lowly detected TcS genes (mean = 37.76%). The orange dot indicates the corresponding mean for frequently detected TcS genes (9.27%).

When analyzing each trypomastigote subpopulation, no coordinated expression among specific TcS members was observed, as no subclusters of cells were identified based on TcS detection profiles. Even when clustering was restricted to genes detected in more than 40% of cells, no clear subclusters of cells were identified (*Figure 4—figure supplement 1b*).

As discussed in the Introduction, although post-transcriptional control remains a central mechanism of gene expression regulation in *T. cruzi*, increasing evidence in recent years has revealed an important contribution of epigenetic mechanisms that modulate gene expression at the transcriptional level. (*Díaz-Viraqué et al., 2023*; *Lima et al., 2021*; *Lima et al., 2022*; *Beati et al., 2023*;

*Ocampo et al., 2025*). Specifically, the loci for TcS genes and other gene families of surface proteins are mostly grouped in specific genomic compartments (*Berná et al., 2018*) that are regulated epigenetically by the activation or silencing of chromatin folding domains (*Díaz-Viraqué et al., 2023*). The TcS that are detected in a high percentage of cells are mostly dispersed throughout the genome (*Supplementary file 4*). This suggests that their preferential expression is likely not due to colocalization in one or a few ubiquitous activated chromatin-folding domains. Nevertheless, mapping the genomic locations of TcS genes detected in a high proportion of cells revealed that most are flanked by core genes (*Figure 5*). The core compartment is enriched in conserved, single-copy genes that typically show more constitutive expression (*Berná et al., 2018*) and, as observed in this study, lower cell-to-cell variability. In contrast, TcS genes that are detected less frequently are preferentially located in the disruptive compartment (*Figure 5*), which is enriched in lineage-specific multigene families and associated with more variable, stage-specific, and potentially stochastic expression under tighter epigenetic control (*Díaz-Viraqué et al., 2023*; *Berná et al., 2018*; *Berná et al., 2017*; *Cruz-Saavedra et al., 2025*). Together, these findings suggest that the higher cellular prevalence of certain TcS transcripts is unlikely to be driven by colocalization within a small number of ubiquitously active chromatin domains but may instead reflect distinct regulatory regimes between the core and disruptive compartments. Future studies integrating single-cell chromatin profiling with scRNA-seq will be required to directly test this model.

## Final remarks

The expression of surface protein-coding genes varied across parasite developmental stages. In particular, genes belonging to the TcS superfamily, which play key roles in host–parasite interactions, exhibited marked heterogeneity in expression among trypomastigotes. Notably, while most TcS genes were detected only in a small fraction of cells, a limited subset was frequently detected across the population. This pattern indicates that TcS expression is not uniform, suggesting the existence of distinct regulatory regimes within the family. Consistent with previous single-cell and bulk RNA-seq studies (*Laidlaw et al., 2025*; *Cruz-Saavedra et al., 2025*), the lack of coordinated expression among TcS members points to a complex regulatory framework that may enable functional diversification among *T. cruzi* subpopulations. Importantly, our analyses indicate that the frequent detection of this TcS subset may be partially explained by their preferential localization within the core genomic compartment, which is associated with more permissive transcriptional environments.

Taken together, our results demonstrate the sensitivity of scRNA-seq for resolving parasite life stages and their associated transcriptional programs, while also pointing to complex regulation of surface protein expression, particularly within the TcS family. Although these results should be considered with caution, as technical limitations inherent to scRNA-seq may influence the observed expression patterns, our interpretation is consistent with all observations presented here and aligns with emerging evidence from independent studies. Collectively, this supports a working hypothesis in which heterogeneous TcS expression may contribute to immune evasion and pathogenicity through a bet-hedging strategy.

## Materials and methods

**Key resources table**

| Reagent type (species) or resource | Designation | Source or reference | Identifiers | Additional information |
|---|---|---|---|---|
| Strain, strain background (*Trypanosoma cruzi*) | Dm28c | *Contreras et al., 1985* | - | - |
| Cell line (*Rattus norvegicus*) | H9c2 | ATCC | CRL-1446 RRID:CVCL_0286 | Provided and validated by ATCC (not verified in-house). |
| Commercial assay, kit | Chromium Next GEM Single Cell 3' | 10x Genomics | v3.1 | Performed by service provider |
| Software, algorithm | Seurat (R) | *Hao et al., 2024* | Version 5 RRID:SCR_016341 | - |
| Software, algorithm | kallisto bustools | *Melsted et al., 2021* | Version 0.51.1 Version 0.45.1 | |
| Software, algorithm | peaks2UTR | *Zhang et al., 2017* | Version 1.2.6 | |

### *Trypanosoma cruzi* and mammalian cell culture

Epimastigote forms of *T. cruzi* strain Dm28c were derived from axenic cultures cultivated in Brain-Heart Infusion medium (BHI, Oxoid) supplemented with 10% heat-inactivated fetal bovine serum (FBS, Capricorn), penicillin (100 units/mL), and streptomycin (100 µg/mL) as described (*Smircich et al., 2023*). Cultures were diluted 1/10 with fresh BHI medium every 3 days and maintained at 28°C.

Myoblast rat cell line H9c2 (provided by ATCC CRL-1446) was maintained in hgDMEM medium (Gibco) supplemented with 10% heat-inactivated fetal bovine serum, penicillin (100 units/mL), and streptomycin (100 µg/mL) at 37°C in a humidified 5% $CO_2$ incubator. H9c2 cell line identity was not independently verified in-house, as the biology of these cells was not the focus of the study. Confluent cells were washed with 1X phosphate-buffered saline (1X PBS), incubated for 5 min with trypsin-EDTA (Gibco), diluted with culture medium, and re-plated for maintenance.

Mycoplasma contamination in cell lines was regularly monitored using MycoAlert Mycoplasma Detection Kit (Lonza), following the manufacturer's protocol.

### Isolation and purification of cellular trypomastigotes and intracellular amastigotes

Late stationary phase epimastigotes of the Dm28c strain were used to infect H9c2 cells for a primary infection. Six days post-infection, cellular trypomastigotes were obtained from the supernatant and were used to infect 50% confluent H9c2 cells at a 10:1 rate. Twenty-four hours post-infection, the cell culture was washed twice with PBS 1X to remove any remaining extracellular parasites and maintained with fresh hgDMEM at 37°C in a humidified 5% $CO_2$ incubator.

For amastigote purification, infected H9c2 cells incubated for 48 hours post-infection were washed with 1X PBS and incubated with trypsin-EDTA for 5 minutes at 37°C. The trypsinization was stopped by adding an equal volume of hgDMEM with 10% FBS. The cell suspension was repeatedly passed through a 27-gauge needle attached to a 30 mL syringe until complete cell disruption was confirmed under the microscope. The supernatant, containing free amastigotes, was collected and centrifuged at 500 x *g* for 10 min at 4°C to remove large host-cell debris. The resulting supernatant was then centrifuged at 4000 x *g* for 10 min at 4°C, and the amastigote-containing pellet was washed twice in chilled 1X DPBS (Dulbecco's Phosphate-Buffered Saline, No Calcium, No Magnesium) and resuspended in 1X DPBS at 200 µL per $1 \times 10^6$ cells, ready for the fixation step.

Cellular trypomastigotes derived from infected H9c2 cells and present in the cell supernatant fraction were collected and centrifuged at 500 x *g* for 10 min at 4°C to remove large host-cell debris. The washing and resuspension steps in DPBS were performed as previously described for amastigotes.

### scRNA-seq library preparation and sequencing

Cell fixation was performed using the Methanol Fixation Protocol for single-cell RNA sequencing (*Gutiérrez-Franco et al., 2023*), after resuspension in DPBS as described above, as recommended by 10X Genomics technical support. Briefly, chilled 100% Methanol (for HPLC, ≥99.9%, Millipore) was added drop by drop ($1 \times 10^6$ cells in 800 µl) and incubated at –20°C for 30 min. For rehydration, fixed cells were first equilibrated at 4°C and then centrifuged at 4000 x *g* for 5 min at 4°C. The supernatant was discarded, and Wash-Resuspension Buffer (3X SSC in Nuclease-free Water, 0.04% UltraPure Bovine Serum Albumin, 1 mM DTT, and 0.2 U/ml RNase Inhibitor) was added to the pellet. Cell debris and large clumps were eliminated by passing the sample through a 40 µm Flowmi Cell Strainer.

10X Genomics library preparation was performed at the service provided by the Instituto de Biología y Medicina Experimental (IBYME, Argentina). The library was sequenced by a service provider (Macrogen, Korea) in a HiSeq2500 equipment (two lanes), generating approximately 880 million reads of 91 bp.

### Transcript quantification

*T. cruzi* 2018 Dm28c genome (*Berná et al., 2018*; release 62, TriTrypDB; *Aslett et al., 2010*) and *T. cruzi* Dm28c maxi circle kDNA sequence (*Berná et al., 2021*) were combined and used as reference. To improve the proportion of reads assigned to genes, 11,362 3'UTR regions of the coding sequences (CDS) were annotated using peaks2UTR (*Haese-Hill et al., 2023*). Gene expression quantification was performed by pseudoalignment using kallisto bustools (*Sullivan et al., 2025*), with the options `--filter` to remove potential noise from environmental RNA and `--em` to apply the

Expectation-Maximization (EM) algorithm. Default values were used for the rest of the parameters. This algorithm outperforms other mapping software in handling multimapping reads, enabling more accurate quantification of multigene families (*Sullivan et al., 2025*). This resulted in 321 million reads mapped to the transcriptome.

## scRNA-seq data processing and analysis

Count matrices from two technical replicates obtained by sequencing the same library were merged: the common barcodes across both datasets were retained, and the count matrices were combined to generate a unified dataset. Subsequently, the following metrics were calculated: nUMI, nGene, and mitoRatio, as well as the log10 of genes per UMI (log10GenesperUMI). A filtering criterion was applied, retaining cells with nUMI >1200, nGene >100, log10GenesperUMI >0.8, and mitoRatio <0.1. Ribosomal rRNA genes were excluded from subsequent analyses. Data normalization and scaling were performed using the Seurat R version 5 package (*Hao et al., 2024*), employing NormalizeData and ScaleData functions.

After quality filtering, 3192 single-cell transcriptomes were retained with an average of 8004 reads mapped per cell. In total, 14,321 genes were detected across all cells (93.5% of the 15,319 annotated protein-coding genes in the *T. cruzi* 2018 Dm28c genome). Per cell, we observed a mean of 1088 detected genes and 2461 UMIs, which corresponds to ~7.1% of the annotated protein-coding gene.

The FindNeighbors function was used to construct a k-nearest neighbors graph of cells using 10 principal components (PCs), and the Louvain algorithm was employed for cell clustering using the FindClusters function. Additionally, doublets were identified and removed using the DoubletFinder R package (*McGinnis et al., 2019*); only 41 doublets were detected, all belonging to cluster 2.

To define marker genes, the FindAllMarkers function from the Seurat package was used, selecting those with an adjusted p-value <0.05 and a log2 fold change (log2FC) >1. To validate these markers, bulk RNA-seq data of Dm28c was incorporated (NCBI BioProject ID PRJNA850400 [24]). Transcripts were quantified using Kallisto (*Bray et al., 2016*; with -b 100 option to perform 100 bootstraps), followed by differential expression analysis conducted with Sleuth (*Pimentel et al., 2017*). Genes with an adjusted p-value <0.05 and |log2FC|>0.25 were filtered for further analysis. Finally, the gene IDs of the Seurat-defined markers were cross-referenced with the IDs of the differentially expressed genes obtained from the bulk RNA-seq analysis to corroborate stage-specific gene expression. For 2D visualization of cell clusters and gene markers expression profiles across cells, UMAP projection was employed (*McInnes et al., 2018*).

Gene IDs corresponding to multigene families (TcS, MASP, Mucins, GP63, RHS, and DGF) were obtained by text searches using the current genome annotation on the 'description' field. Throughout the manuscript, these genes are referred to as either surface protein genes or multigene family genes. Single and low copy number genes were defined as those that did not belong to the latter gene list. For simplicity, 'single-copy' will be used to refer to these genes throughout the manuscript. To avoid biases arising from differences in list size between single-copy and multigene family gene sets, we generated a subsampled single-copy gene list by randomly selecting the same number of genes as in the multigene family set. The expression distribution of this subsampled single-copy gene set is similar to that of the full dataset.

Data processing was conducted using R version 4.2.0. Statistical analyses were performed using the Wilcoxon rank-sum test and p-values <0.05 were considered statistically significant.

To study expression inequality, we plotted Lorenz curves (*Lorenz, 1905*) and applied the Gini index, a metric originally developed in the field of economics (*Atkinson, 1970*). Both metrics serve as a measure of expression heterogeneity, and we employed it in two distinct contexts: first, to evaluate the degree of inequality in the distribution of a gene's expression levels across individual cells (*Figure 2d*, *Figure 2—figure supplement 1d* and *Supplementary file 3*); and second, to assess the extent to which a given cell expresses individual genes at varying rates (*Figure 4c*).

TcS genomic localization analysis was performed by defining directional gene clusters from the GFF annotation: protein-coding genes were ordered by coordinate per contig and consecutive genes on the same strand were grouped into a DGC, with a strand switch (change from '+' to '-' or vice versa) marking DGC boundaries. For each gene, we calculated the percentage of its polycistronic neighbors (genes within the same DGC) that belong to the multigene family gene set previously defined and compared high-abundance TcS vs low-abundance TcS (sampling n=30). For statistical comparisons,

we assessed the robustness of the analysis by sampling 50 iterations of n=30 low-abundance TcS, and distributions were compared to the high-abundance set using two-sided Wilcoxon rank-sum tests with Benjamini-Hochberg correction.

## Acknowledgements

This project was supported by: CSIC, Universidad de la República, grant number: I+D-2020-505 awarded to PS and UNSAM INVESTIGA 2025, grant number: 80020250100103SM to JGDG; LI, LB, JG, MD, JSS and PS received financial support from PEDECIBA, and VAC and JGDG are members of the Research Career of CONICET. The funders had no role in study design, data collection and analysis, decision to publish, or preparation of the manuscript.

## Additional information

### Funding

| Funder | Grant reference number | Author |
| --- | --- | --- |
| CSIC, Universidad de la Republica | I+D-2020-505 | Pablo Smircich |
| Programa de Desarrollo de las Ciencias Básicas | | Lucas Inchausti<br>Lucia Bilbao<br>Joaquín Garat<br>José Sotelo-Silveira<br>Maria A Duhagon<br>Pablo Smircich |
| UNSAM INVESTIGA 2025 | 80020250100103SM | Javier G De Gaudenzi |

The funders had no role in study design, data collection and interpretation, or the decision to submit the work for publication.

### Author contributions

Lucas Inchausti, Formal analysis, Validation, Investigation, Visualization, Methodology, Writing – original draft, Writing – review and editing; Lucia Bilbao, Vanina A Campo, Investigation, Methodology, Writing – review and editing; Joaquín Garat, Formal analysis, Methodology, Writing – review and editing; José Sotelo-Silveira, Maria A Duhagon, Conceptualization, Resources, Writing – review and editing; Gabriel Rinaldi, Virginia M Howick, Conceptualization, Methodology, Writing – review and editing; Javier G De Gaudenzi, Conceptualization, Resources, Investigation, Methodology, Writing – original draft, Writing – review and editing; Pablo Smircich, Conceptualization, Resources, Supervision, Funding acquisition, Validation, Investigation, Methodology, Writing – original draft, Project administration, Writing – review and editing

### Author ORCIDs

Lucas Inchausti ⓘ https://orcid.org/0009-0009-7089-6213
Lucia Bilbao ⓘ https://orcid.org/0009-0009-8895-1516
Vanina A Campo ⓘ https://orcid.org/0000-0003-2096-5988
Joaquín Garat ⓘ https://orcid.org/0009-0005-5505-8036
José Sotelo-Silveira ⓘ https://orcid.org/0000-0002-4758-8556
Gabriel Rinaldi ⓘ https://orcid.org/0000-0002-7767-4922
Virginia M Howick ⓘ https://orcid.org/0000-0003-3054-2460
Maria A Duhagon ⓘ https://orcid.org/0000-0001-6468-1704
Javier G De Gaudenzi ⓘ https://orcid.org/0000-0001-8349-8833
Pablo Smircich ⓘ https://orcid.org/0000-0002-3856-3920

Reviewer #1 (Public review): https://doi.org/10.7554/eLife.105822.3.sa1
Reviewer #2 (Public review): https://doi.org/10.7554/eLife.105822.3.sa2
Reviewer #3 (Public review): https://doi.org/10.7554/eLife.105822.3.sa3
Author response https://doi.org/10.7554/eLife.105822.3.sa4

# Additional files

## Supplementary files

Supplementary file 1. Marker genes for identified cell clusters by scRNA-seq. (a) Genes significantly upregulated in cluster 0 (adjusted *P*-value <0.05), (b) Genes significantly upregulated in cluster 1 (adjusted *P*-value <0.05) and (c) Genes significantly upregulated in cluster 2 (adjusted *P*-value <0.05)

Supplementary file 2. Correlation of results from differential expression analysis using bulk RNA-seq data with the top 10 gene markers from each of the three identified clusters from scRNA-seq data analysis. Positive log2FC corresponds to upregulated genes in the trypomastigote stage, whereas negative log2FC corresponds to upregulated genes in the amastigote stage. Gene C4B63_196g28 showed no statistically significant changes in expression. Genes without values were not identified as differentially expressed in the bulk RNA-seq data.

Supplementary file 3. Statistical analysis results from gene expression heterogeneity of gene groups. (a) Summary of the number of cells expressing genes by gene group. Shown are the mean, relative standard deviation (rsd), and median of the number of cells expressing each gene within the indicated groups. (b) Pairwise statistical comparisons between gene groups based on the number of cells expressing each group. Differences between groups were assessed using the Wilcoxon rank-sum test. (c) Summary statistics of the Gini index across gene groups. The Gini index was computed for each gene as a quantitative measure of expression inequality across cells, derived from the corresponding Lorenz curves from *Figure 2d*. Reported values indicate the mean, relative standard deviation (rsd), and median Gini index for each gene group. (d) Pairwise statistical comparisons of Gini index distributions between gene groups. Differences between groups were assessed using the Wilcoxon rank-sum test. Reported values correspond to p-values for each pairwise comparison.

Supplementary file 4. Percentage of cells in which each TcS is expressed in trypomastigote cell population. (a) All trypomastigote cells, (b) Trypomastigote cells corresponding to the trypo_ 0 cluster and (c) Trypomastigote cells corresponding to the trypo_ 1 cluster.

MDAR checklist

## Data availability

Raw sequence data is available in the SRA (https://www.ncbi.nlm.nih.gov/sra/) under BioProject PRJNA1200704. The code for the analyses presented in this paper is openly accessible at https://github.com/bioinfo-iibce/scRNAseq_Tcruzi (copy archived at *bioinfo-iibce, 2026*).

The following dataset was generated:

| Author(s) | Year | Dataset title | Dataset URL | Database and Identifier |
| --- | --- | --- | --- | --- |
| Instituto de Investigaciones Biologicas Clemente Estable | 2024 | Single-Cell RNA-Seq Investigation of Gene Expression Dynamics in Trypanosoma cruzi Populations | https://www.ncbi.nlm.nih.gov/bioproject/PRJNA1200704 | NCBI BioProject, PRJNA1200704 |

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
