## [Editor Report · eLife Assessment]

This study provides **important** insights into how *Trypanosoma cruzi* populations diversify surface protein expression, showing through single-cell RNA sequencing that trans-sialidase-like genes are expressed heterogeneously across individual parasites, a pattern with clear implications for immune evasion. The evidence is **convincing**, supported by robust single-cell transcriptomic analyses, consistent quantitative measures of expression heterogeneity, and integration with genomic organization that together argue against purely stochastic expression.

---

## [Referee Report · Reviewer #1 (Public review)]

Summary:

The authors aimed to assess the variability in expression of surface protein multigene families between amastigote and trypomastigote *Trypanosoma cruzi*, as well as between individuals within each population. The analysis presented shows higher expression of multigene family transcripts in trypomastigotes compared to amastigotes and that there is variation in which copies are expressed between individual parasites. Notably, they find no clear subpopulations expressing previously characterised trans-sialidase groups and that no patterns of coexpressed TcS genes were evident within individual cells or subpopulations. They also note that TcS encoded in the core genome are more often expressed, compared to TcS genes encoded in other genome compartments.

Strengths:

Additionally, the authors successfully process methanol fixed parasites with the 10x Genomics platform. This approach is valuable for other studies where using live parasites for these methods is logistically challenging.

In this second submission the authors show the kallisto mapping approach used is as robust as possible, and that this approach outperforms STAR mapping.

Weaknesses:

The authors describe a single experiment, which lacks repeats, controls or complementation with other approaches and the investigation is limited to the trans-sialidase transcripts.

**Comments on revised version:**

Thank you to the authors for taking the time to thoroughly address the peer review. The main concerns have now been addressed, and the manuscript edited to make points of confusion clearer.

---

## [Referee Report · Reviewer #2 (Public review)]

Summary:

This manuscript presents a valuable single-cell RNA-seq study on *Trypanosoma cruzi*, an important human parasite. It investigates the expression heterogeneity of surface proteins, particularly those from the trans-sialidase-like (TcS) superfamily, within amastigote and trypomastigote populations. The findings suggest a previously underappreciated level of diversity in TcS expression, which could have implications for understanding parasite-host interactions and immune evasion strategies. The use of single cell approaches to delve into population heterogeneity is strong. However, the study does have some limitations that need to be addressed.

The focus on single-cell transcriptional heterogeneity in surface proteins, especially the TcS family, in *T. cruzi* is novel. Given the important role of these proteins in parasite biology and host interaction, the findings have potential significance.

Strengths:

The key finding of heterogeneous TcS expression in trypomastigotes is well-supported. The analysis comparing multigene families, single-copy genes, and ribosomal proteins highlights the unusual nature of the variation in surface protein coding genes.

Weaknesses:

While the manuscript identifies TcS heterogeneity, the functional implications of the different expression profiles remain speculative. The authors state it may reflect differences in infectivity, but no direct experimental evidence supports this.

The manuscript lacks any functional validation of the single-cell findings. For instance, do the trypomastigote subpopulations identified based on TcS expression exhibit differences in infectivity, host cell tropism, or immune evasion? Such experiments would greatly strengthen the study.

The authors identify a subpopulation of TcS genes that are highly expressed in many cells. However, it is unclear if these correspond to previously characterized TcS members with specific functions.

The authors hypothesize that observed heterogeneity may relate to chromatin regulation. However, the study does not directly address these mechanisms. There are interesting connections to be made with what they identify as colocalization of genes within chromatin folding domains, but the authors do not fully explore this. It would be insightful to address these mechanisms in future work. [...]

**Comments on revisions:**

The novel version of the manuscript has improved and satisfied this reviewer.

---

## [Referee Report · Reviewer #3 (Public review)]

The study aimed to address a fundamental question in *T. cruzi* and Chagas disease biology - how much variation is there in gene expression between individual parasites? This is particularly important with respect to the surface protein-encoding genes, which are mainly from massive repetitive gene families with 100s to 1000s of variant sequences in the genome. There is very little direct evidence for how expression of these genes is controlled. The authors conducted a single cell RNAseq experiment of in vitro cultured parasites with a mixture of amastigotes and trypomastigotes. Most of the analysis focused on the heterogeneity of gene expression patterns amongst trypomastigotes. They show that heterogeneity was very high for all gene classes, but surface-protein encoding genes were the most variable. Interestingly, in the case of the trans-sialidase genes, many sequence variants were detected in fewer than 5% of parasites while a subset of 31 others was detected in >40% if parasites, hinting at compartmentalised expression control within the gene family. The biology of the parasite (e.g. extensive post-transcriptional regulation) and potential technical caveats (e.g. high dropout rates across the genome) make it difficult to infer connections to actual protein expression on the parasite surface, but the results are a significant advance for the field.

(1) Limit of detection and gene dropouts.

An average of ~1100 genes are detected per parasite which indicates a dropout rate of over 90%. It appears that RNA for the "average" single copy 'core' gene is only detected in around 3% of the parasites sampled (Figure 2c: ~100 / 3192). While comparable with some other trypanosome scRNAseq studies, this remains a caveat to the interpretation that high cell-to-cell variability in gene expression is explained by biological factors. The argument would be more convincing if the dropout rates and expression heterogeneity were minimal for highly expressed housekeeping genes. The authors are appropriately cautious in their interpretation and acknowledge the need for further validation.

(2) Heterogeneity across the board.

The authors focus on the relative heterogeneity in RNA abundance for surface proteins from the multicopy gene families vs core genes. While multicopy gene sequences do show significantly more cell-to-cell variability, there is still surprisingly high inequality of expression amongst genes in other classes including single copy housekeeping and ribosomal genes. Again the biological relevance of the comparison is uncertain and the authors acknowledge the need for further investigation.

This study provides some tantalising evidence that the expression of surface genes may vary substantially between individual parasites in a single clonal population. The study is also amongst the very first to apply scRNAseq to *T. cruzi*, so the broader data set will be an important resource for researchers in the field.

**Comment on revised version:**

The manuscript is significantly improved. The revised explanations and figures make several aspects of the data analysis and interpretation much clearer to me now. Thanks to the authors.

---

## [Author Response]

The following is the authors’ response to the original reviews

**Public Reviews:**

**Reviewer #1 (Public review):**
Summary:The authors aimed to assess the variability in the expression of surface protein multigene families between amastigote and trypomastigote *Trypanosoma cruzi*, as well as between individuals within each population. The analysis presented shows higher expression of multigene family transcripts in trypomastigotes compared to amastigotes and that there is variation in which copies are expressed between individual parasites. Notably, they find no clear subpopulations expressing previously characterised trans-sialidase groups. The mapping accuracy to these multicopy genes requires demonstration to confirm this, and the analysis could be extended further to probe the features of the top expressed genes and the other multigene families also identified as variable.Strengths:The authors successfully process methanol-fixed parasites with the 10x Genomics platform. This approach is valuable for other studies where using live parasites for these methods is logistically challenging.Weaknesses:The authors describe a single experiment, which lacks controls or complementation with other approaches and the investigation is limited to the trans-sialidase transcripts.It would be more convincing to show either bioinformatically or by carrying out a controlled experiment, that the sequencing generated has been mapped accurately to different members of multigene families to distinguish their expression. If mapping to the multigene families is inaccurate, this will impact the transcript counts and downstream analysis.

We thank the reviewer for raising these important points.

We agree that the analysis of multigene families at the single-cell level is an important question, particularly given the heterogeneity observed across several of them. However, the aim of this short report is not to provide a comprehensive analysis of the entire experiment, but rather to focus on what we consider an important biological phenomenon observed in TcTS genes.

Regarding the mapping accuracy of the reads, we acknowledge that this can limit the disambiguation of highly similar multicopy transcripts. This is, in fact, a common challenge when analyzing transcriptomic data from *T. cruzi*.

To address this issue, we analyzed the sequence identity of the 3′ ends of TcS transcripts (defined as the 3′UTR plus 20% of the CDS region). As shown in Author response image 1, these regions display a median sequence identity of approximately 25%, indicating that sufficient sequence divergence exists for mapping algorithms to use during read assignment.

In addition, it is important to note that kallisto, the software used in our analysis, was specifically designed to address multimapping reads through pseudoalignment combined with an expectation-maximization algorithm that probabilistically assigns reads across compatible transcripts.

To directly assess performance, we simulated reads from the *T. cruzi* transcriptome used in this study (3′UTRs plus 20% of the CDS regions) and compared two mapping/counting strategies: (a) transcriptome pseudoalignment using kallisto, and (b) genome alignment followed by counting using STAR + featureCounts. The latter approximates the strategy implemented in CellRanger, the standard pipeline for quantifying expression levels from 10X Genomics single cell RNA-seq data. We found that kallisto recovered the simulated “true” counts with substantially higher accuracy than STAR + featureCounts (Pearson correlation: all genes, 0.991 vs 0.595; surface protein genes, 0.9996 vs 0.827; trans-sialidase (TcS) genes, 0.9998 vs 0.773). These results indicate that pseudoalignment is currently the optimal strategy for recovering the relative expression of highly similar gene family members (Author response image 1 C).

**Author response image 1. sa4fig1:** (A) Distribution of pairwise sequence identity values calculated among the 3′-end regions of all transcripts (defined as the 3′UTR plus 20% of the coding sequence). (B) Distribution of read mapping coordinates over all multigene family transcripts normalized as percentage of the gene length (C) Scatter plots showing the correlation between estimated transcript counts obtained using kallisto (red) and STAR + featureCounts (grey) versus the corresponding simulated ground-truth values.

**Reviewer #2 (Public review):**
Summary:This manuscript presents a valuable single-cell RNA-seq study on *Trypanosoma cruzi*, an important human parasite. It investigates the expression heterogeneity of surface proteins, particularly those from the trans-sialidase-like (TcS) superfamily, within amastigote and trypomastigote populations. The findings suggest a previously underappreciated level of diversity in TcS expression, which could have implications for understanding parasite-host interactions and immune evasion strategies. The use of single-cell approaches to delve into population heterogeneity is strong. However, the study does have some limitations that need to be addressed.The focus on single-cell transcriptional heterogeneity in surface proteins, especially the TcS family, in *T. cruzi* is novel. Given the important role of these proteins in parasite biology and host interaction, the findings have potential significance.Strengths:The key finding of heterogeneous TcS expression in trypomastigotes is well-supported. The analysis comparing multigene families, single-copy genes, and ribosomal proteins highlights the unusual nature of the variation in surface protein-coding genes.Weaknesses:While the manuscript identifies TcS heterogeneity, the functional implications of the different expression profiles remain speculative. The authors state it may reflect differences in infectivity, but no direct experimental evidence supports this.The manuscript lacks any functional validation of the single-cell findings. For instance, do the trypomastigote subpopulations identified based on TcS expression exhibit differences in infectivity, host cell tropism, or immune evasion? Such experiments would greatly strengthen the study.

We thank the reviewer for their careful reading of the manuscript. We agree that obtaining experimental evidence on the influence of multiple multigene families would represent a significant advancement in the field. However, we would like to emphasize that this study is presented as a short communication centered on a specific and biologically relevant observation within a single multigene family. The aim of the manuscript is to highlight what we consider an important biological phenomenon that raises hypotheses to be tested in future work.

The influence of phenotypic heterogeneity and its possible advantages under environmental pressures has been previously proposed for *Trypanosoma cruzi*, related trypanosomatids, and other biological systems, ranging from bacteria to tumors (Seco-Hidalgo 2015, doi: 10.1098/rsob.150190 and Luzak 2021, doi: 10.1146/annurev-micro-040821-012953, for a comprehensive review on this topic). While the reviewer is correct in noting that our model does not demonstrate a functional role for TcTS heterogeneity, the experimental approaches required to address this question in a large multigene family are highly complex. This is particularly challenging in *T. cruzi*, where the study of multigene families is limited by the restricted set of available molecular biology tools (such as RNAi). Therefore, further experimental validation of these observations falls outside the scope of this short report.

In this revised version, we have included additional validation and clarification of the results, as well as a more explicit discussion of their limitations. In addition, we present a preliminary analysis exploring potential mechanisms that could coordinate the observed expression patterns of the TcTS family.

The authors identify a subpopulation of TcS genes that are highly expressed in many cells. However, it is unclear if these correspond to previously characterized TcS members with specific functions.

The TcS subgroup with a high frequency of detection comprises 31 genes, none of which belong to the catalytically active Group I trans-sialidases. Instead, this subgroup includes members of Groups II, III, IV, V, VI, and VIII. This information has been added to Supplementary Table 3 and is now stated in the revised manuscript.

The authors hypothesize that observed heterogeneity may relate to chromatin regulation. However, the study does not directly address these mechanisms. There are interesting connections to be made with what they identify as the colocalization of genes within chromatin folding domains, but the authors do not fully explore this. It would be insightful to address these mechanisms in future work.

In response to the reviewer’s and editorial team’s request for additional mechanistic insight into the regulatory processes that may be involved in the observed patterns, we have expanded the revised manuscript to discuss how the genomic context of TcS loci could contribute to the observed heterogeneity in TcS expression. As noted in the original version of the manuscript, TcS genes and other surface-protein gene families are largely partitioned into discrete genomic compartments, whose expression has been reported to be regulated by epigenetic control of chromatin-folding domains (doi.org/10.1038/s41564-023-01483-y). However, we previously showed that TcS genes detected in a high proportion of cells are, in most cases, dispersed throughout the genome, arguing against a model in which their preferential expression results from colocalization within a small number of ubiquitously activated chromatin domains. In response to the reviewer’s suggestion, we performed a more detailed analysis of the genomic locations of these TcS genes. We found that many of them are localized within the core compartment (new Figure 5). Because the core compartment is enriched for conserved, housekeeping genes that typically display more constitutive expression (doi.org/10.1038/s41564-023-01483-y), whereas the disruptive compartment is enriched for lineage-specific multigene families associated with variable, stage-specific, and recently reported stochastic expression (doi.org/10.1038/s41467-025-64900-2), our results are consistent with a model in which compartment-specific regulatory mechanisms (in addition to post-transcriptional regulation) influence the differential cellular expression of core- versus disruptive-located TcS genes. We have incorporated these results and discussion in the revised manuscript.

The merging of technical replicates needs further justification and explanation as they were not processed through separate experimental conditions. While barcodes were retained, it would be informative to know how well each technical replicate corresponds with the other. If both datasets were sequenced on the same lane, the inclusion of technical replicates adds noise to the analysis.

Regarding technical details, we now include the total number of mapped reads and average number of reads mapped per cell new paragraph in the Methods section.

The technical replicates consist of a single Illumina library that was sequenced in two separate runs. As this approach is expected to be highly reproducible, we merged both runs into a single count table. To support this decision, we assessed the concordance between the two sequencing runs and observed an almost perfect correlation between them (Author response image 2).

**Author response image 2. sa4fig2:** Correlation analysis of number of reads assigned to cells between technical replicate 1 and technical replicate 2.

While the number of cells sequenced (3192) seems reasonable, it's not clear how much the conclusions are affected by the depth of sequencing. A more detailed description of the sequencing depth and its impact on gene detection would be valuable.

We detected a mean of 1088 genes per cell. Based on the 15,319 annotated protein-coding genes in the reference genome, this represents 7.1% of the *T. cruzi* protein-coding gene complement detected in each cell.

Across the entire dataset, a total of 14,321 genes were detected in at least one cell, representing 93.5% of all annotated protein-coding genes. This suggests that our experiment captured a broad representation of the parasite's transcriptome.

This per-cell detection rate is characteristic of droplet-based scRNA-seq and is consistent with other trypanosomatid studies. For example, the *T. brucei* single-cell atlas (Hutchinson et al., 2021) reported a median detection of 1052 genes per cell. In the case of *T. cruzi*, the recently published pre-print of the *T. cruzi* single cell atlas from Laidlaw & García-Sánchez et al. reported a mean between 298 and 928 genes detected per cell (depending on the sample).

This information is now included in Methods.

While most of the methods are clear, the way in which the subsampled gene lists were generated could be more thoroughly described, as some details are not clear for the subsampling of single-copy genes.

The subsampling method was originally described in the Figure 2 legend; to better highlight this approach, we have now moved its description to the Methods section.

Some of the figures are difficult to interpret. For example, the color scaling in the heatmap of Supplementary Figure 3B is not self-explanatory and it is hard to extract meaningful conclusions from the graph.

We agree with the reviewer in this assessment. We have now modified the figures to be more self-explanatory and better reflect the conclusions.

**Reviewer #3 (Public review):**
The study aimed to address a fundamental question in *T. cruzi* and Chagas disease biology - how much variation is there in gene expression between individual parasites? This is particularly important with respect to the surface protein-encoding genes, which are mainly from massive repetitive gene families with 100s to 1000s of variant sequences in the genome. There is very little direct evidence for how the expression of these genes is controlled. The authors conducted a single-cell RNAseq experiment of in vitro cultured parasites with a mixture of amastigotes and trypomastigotes. Most of the analysis focused on the heterogeneity of gene expression patterns amongst trypomastigotes. They show that heterogeneity was very high for all gene classes, but surface-protein encoding genes were the most variable. In the case of the trans-sialidase gene family, many sequence variants were only detected in a small minority of parasites. The biology of the parasite (e.g. extensive post-transcriptional regulation) and potential technical caveats (e.g. high dropout rates across the genome) make it difficult to infer what this might mean for actual protein expression on the parasite surface.

We thank the reviewer for this important comment, highlighting a central challenge when studying trypanosomatid biology. We acknowledge that in most eukaryotes and particularly in *T. cruzi*, where there is a predominant role of post-transcriptional regulation, mRNA levels are not always directly correlated with protein abundance, as previously reported by us and others (10.1186/s12864-015-1563-8, 10.1128/msphere.00366-21, 10.1590/S0074-02762011000300002, 10.1042/bse0510031). Nevertheless, steady-state transcript levels obtained by RNA-seq remain informative for assessing differential gene expression, and this approach has been widely used as a proxy for the study of gene expression profiles in *T. cruzi* (10.7717/peerj.3017, 10.1371/journal.ppat.1005511, 10.1016/j.jbc.2023.104623, 10.3389/fcimb.2023.1138456, 10.1186/s13071-023-05775-4).

It's also interesting to note that recent proteomic analyses (10.1038/s41467-025-64900-2) have revealed substantial heterogeneity in the expression of surface proteins, including trans-sialidases, supporting the idea that the transcriptional heterogeneity we observe reflects a genuine biological feature that propagates to the protein level.

We have now added a sentence to the discussion acknowledging this limitation and discussed the results from Cruz-Saavedra, et al. in the revised manuscript.

(1) Limit of detection and gene dropoutsAn average of ~1100 genes are detected per parasite which indicates a dropout rate of over 90%. It appears that RNA for the "average" single copy 'core' gene is only detected in around 3% of the parasites sampled (Figure 2c: ~100 / 3192). This may be comparable with some other trypanosome scRNAseq studies, but this still seems to be a major caveat to the interpretation that high cell-to-cell variability in gene expression is explained by biological rather than technical factors. The argument would be more convincing if the dropout rates and expression heterogeneity were minimal for well-known highly expressed genes e.g. tubulin, GAPDH, and ribosomal RNAs. Admittedly, in their Final Remarks, the authors are very cautious in their interpretation, but it would be good to see a more thorough discussion of technical factors that might explain the low detection rates and how these could be tested or overcome in future work.(2) Heterogeneity across the boardThe authors focus on the relative heterogeneity in RNA abundance for surface proteins from the multicopy gene families vs core genes. While multicopy gene sequences do show more cell-to-cell variability, the differences (Figure 2D) are roughly average Gini values of 0.99 vs 0.97 (single copy) or 0.95 (ribosomal). Other studies that have applied similar approaches in other systems describe Gini values of < 0.2-0.25 for evenly expressed "housekeeping" genes (PMIDs 29428416, 31784565). Values observed here of >0.9 indicate that the distribution for all gene classes is extremely skewed and so the biological relevance of the comparison is uncertain.

We recognize the limitations imposed by gene dropout in our data, as highlighted by the reviewer. Unfortunately, gene dropout is an inherent limitation of 10x genomics data. Trypanosomatids are not an exception in this regard, and the general metrics of the single-cell RNA-seq data in other reports are equivalent to those obtained in our experiment.

Despite this important limitation, we believe that our comparative analyses (the contrast between TcS and ribosomal protein expression) provide valuable insights into a biological phenomenon with potential functional relevance for the parasite. Furthermore, we are actively working on generating single-cell RNA-seq data using alternative methodologies that improve gene dropout rates. We anticipate that these future studies will help clarify the extent of the phenomenon described in this work.

Our results reveal a small subset of TcS genes that are frequently detected across cells, a pattern that is not compatible with random detection unless these genes were highly expressed and preferentially captured by random sampling. However, as shown in Figure 4b, many genes expressed at comparable levels are not detected at high frequencies. In line with this, Figure 4c shows that within individual cells, the detected TcS genes exhibit similar expression levels. Finally, we confirmed that this frequently detected subset shows high read counts at the bulk RNA-seq level (Figure 4 - Figure Supplement 1), consistent with the fact that these TcS are frequent in the population even when they are not specially highly expressed within each cell. Taken together, these findings argue against a purely random sampling of TcS genes and support the interpretation that this pattern reflects an underlying biological feature. We agree that further validation will be required. Accordingly, since the initial submission, we have been careful to frame our conclusions conservatively, explicitly noting that dropout remains a limitation of these data that could influence the observed patterns. In the revised version, we have strengthened this point by including a specific statement in the final remarks. Our interpretation is presented as a working hypothesis that is fully compatible with the observations reported here and may be informative for the field. To better reflect this reasoning, we have revised Figure 4b, expanded the discussion, and explicitly included this limitation in the final remarks of the revised manuscript.

Nevertheless, this study does provide some tantalising evidence that the expression of surface genes may vary substantially between individual parasites in a single clonal population. The study is also amongst the very first to apply scRNAseq to *T. cruzi*, so the broader data set will be an important resource for researchers in the field.

We thank the reviewer for highlighting the relevance of our study and for their positive assessment of the potential significance of these observations. We also agree that the dataset generated here may represent a useful resource for the community.

**Recommendations for the authors:**

**Reviewer #1 (Recommendations for the authors):**
(1) In Figures 1c and 1d, it would be useful to include the genes as the plot titles.

We agree with the reviewer that including gene names in the plot makes the panels more self-explanatory. We have added gene names to the updated version of Figure 1.

(2) Can you include the read lengths of the sequencing and whether this is sufficient to map accurately to very similar genes of the same multigene family? As stated in the public summary, this would make the data far more convincing as standard 10x chromium cannot distinguish similar gene copies unless a longer read 2 is used. Given that only the 3' end is targeted, is this enough to distinguish the TcS and other mutligene family transcripts?

We thank the reviewer for raising this important point. We agree that short 3′ biased reads can limit the disambiguation of highly similar multicopy transcripts. This is, in fact, a common challenge when analyzing transcriptomic data from *T. cruzi*.

To address this issue, we analyzed the sequence identity of the 3′ ends of TcS transcripts (defined as the 3′UTR plus 20% of the CDS region). As shown in Author response image 1, these regions display a median sequence identity of approximately 25%, indicating that sufficient sequence divergence exists for mapping algorithms to use during read assignment.

In addition, it is important to note that kallisto, the software used in our analysis, was specifically designed to address multimapping reads through pseudoalignment combined with an expectation-maximization algorithm that probabilistically assigns reads across compatible transcripts.

To directly assess performance, we simulated reads from the *T. cruzi* transcriptome used in this study (3′UTRs plus 20% of the CDS regions) and compared two mapping/counting strategies: (a) transcriptome pseudoalignment using kallisto, and (b) genome alignment followed by counting using STAR + featureCounts. The latter approximates the strategy implemented in CellRanger, the standard pipeline for quantifying expression levels from 10X Genomics single cell RNA-seq data. We found that kallisto recovered the simulated “true” counts with substantially higher accuracy than STAR + featureCounts (Pearson correlation: all genes, 0.991 vs 0.595; surface protein genes, 0.9996 vs 0.827; trans-sialidase (TcS) genes, 0.9998 vs 0.773). These results indicate that pseudoalignment is currently the optimal strategy for recovering the relative expression of highly similar gene family members (Author response image 1C).

The length of the R2 read (91bp) was included in Methods (line 411).

(3) It is stated that 'single copy' genes also include 'low copy number genes". What does this include exactly? Is it more actuate to say non-surface protein genes?

The distinction we aim to make is between multigene families and the rest of the genome. Most multigene families encode surface proteins, but not all surface protein genes belong to multigene families. To clarify this point we included a sentence in methods to reflect that when we describe “surface proteins” we are referring to surface proteins coded by multigene families (line 453). In addition, long-read genomic DNA sequencing and assembly have revealed that many genes previously believed to be single-copy are actually duplicated at low copy numbers (doi.org/10.1099/mgen.0.000177). For this reason, we extend the concept of “single-copy” genes to include those that have only a few duplicates.

(4) It is stated in line 127 that TcS have particular high heterogeneity - it does not look that way by eye compared to the other multigene families. Can statistic be used to prove this, or simply state the decision was made to focus on the TcS?

As noticed by the reviewer, all multigene families show significantly higher heterogeneity compared to single-copy genes, as stated in the text and shown in figure legends from Figure 2, Supplementary Figure 1 and the new Supplementary Table 2.

That said, it was not the statistical results that guided our decision to focus on TcS, but rather their well-established biological relevance in *T. cruzi*. As suggested, we have now emphasized this rationale more clearly in the revised text (lines 160-167).

Besides, recent work has shown that TcS genes exhibit a bimodal distribution of expression levels using bulk RNA-seq data, in contrast to core genes and other multigene families (doi.org/10.1038/s41467-025-64900-2, doi.org/10.1038/s41564-023-01483-y). This distinct regulatory behavior further justifies our decision to examine TcS separately.

(5) Expression of different TcS has been investigated between the different life cycle stages for a few individual genes previously (Freitas et al). Can the authors not extend this investigation to all the genes detect by scRNA-seq here to demonstrate those with higher/lower expression in amastigotes vs trypomastigotes building on Figure 2A? Are particular groups linked to either stage?

We performed this analysis and did not observe any correlation between TcS groups and life cycle stage. In all cases TcS were more frequently detected in trypomastigotes. This difference was statistically significant for all groups except group VII, likely due to the low number of genes analyzed in this group (Author response image 3).

**Author response image 3. sa4fig3:** Per-gene number of expressing cells by TcS group and life-stage. Boxplots show, for each TcS group (I–VIII), the distribution across genes of the number of cells in which the gene is detected. Each point represents a single TcS; Amastigote cells: green points/boxes, Trypomastigote cells: salmon points/boxes. The y-axis is on log10 scale. Asterisks indicate statistically significant differences from the comparison between Amastigote and Trypomastigote within each TcS group, assessed using a paired two-sided Wilcoxon signed-rank test: * p < 0.05, ** p < 0.01, *** p < 0.001.

(6) What exactly is the Z-score shown in Figure 2B?

In this analysis num_multigene represents the number of multigene family genes detected in each individual cell. For every cell, we counted how many genes from our predefined multigene family gene list has detectable expression (more than zero UMI counts); in the UMAP plot, this value is reflected by the size of each point. On the other hand, z_multigene captures the relative expression level of multigene family genes within each cell. This metric is calculated by summing the UMI counts of all multigene family genes per cell and then standardizing this value across the dataset using a z-score transformation, such that positive values reflect above-average multigene family expression and negative values reflect below-average levels. In the UMAP plot, this metric determines the color scale of each point. Taking together num_multigene and z_multigene allow us to distinguish cells that express multigene family genes broadly (high gene counts), strongly (high relative expression), both, or neither, and to relate these patterns to identified cell populations.

We included a short description in legend of the new version of Figure 2 (lines 176-180).

(7) For the reclustering of trypomastigotes based on TcS genes alone, please show the UMAP and discuss why the resolution giving two clusters is chosen? I assume increasing the resolution does not reveal clusters of cells express one of the 8 groups of TcS for example?

We appreciate the reviewer’s suggestion. In this analysis, our goal was to test whether the phenotypic heterogeneity previously reported in trypomastigotes could be recapitulated using TcS genes alone, as prior studies described two major transcriptomic phenotypes within this stage.

Increasing the clustering resolution did not reveal subclusters corresponding to the eight TcS sequence groups. This might reflect the fact that these groups are defined based on sequence similarity rather than on expression patterns, as noted by Freitas et al. (doi:10.1371/journal.pone.0025914).

(8) In Figure 4B, there may be an upward trend in the level of expression and the number of cells a transcript is detected in? It would be worth showing this is or is not the case with statistics if possible.

The number of genes detected in a high proportion of cells is low, which limits the statistical power of this analysis. Also, substantial dispersion is observed within the 0-5% interval. Nevertheless, this figure is presented primarily to highlight that a considerable number of highly expressed genes are detected in only a small fraction of cells. If expression level were the main determinant of detection frequency across cells, one would expect very few highly expressed genes to fall within the 0-5% interval. Contrary to this expectation, among the 50 highest expressed TcS genes, 62% are detected in fewer than 5% of cells, and even among the top 10 most highly expressed TcS genes, 40% fall within this lowest detection group. To facilitate this interpretation, we modified the figure (new Figure 4b) to explicitly highlight the top 50 most expressed TcS genes and incorporated this discussion into the main text of the revised manuscript (lines 244-251), making the conclusion clearer to the reader.

(9) Do the cells group instead by expression of any of the other multigene families not investigated in detail?

It is possible that additional transcriptional substructure among trypomastigotes is driven by the expression of other multigene families beyond TcS. In this short report (with limited number of figures, words, etc.), we focused specifically on the trans-sialidase family as discussed earlier. A more comprehensive analysis including other large surface gene families (MASPs, mucins, GP63) is planned as part of ongoing work and will be presented in future reports.

**Reviewer #2 (Recommendations for the authors):**
This reviewer suggests the conduction of functional experiments in follow-up studies to establish links between TcS expression profiles and parasite behavior and into potential regulatory mechanisms responsible for the observed TcS heterogeneity, particularly focusing on epigenetic modifications. It would be interesting to correlate the highly expressed TcS members identified here with previously characterized TcS isoforms and provide more description regarding which particular groups and TcS members are driving the findings. It would benefit from further clarification regarding sequencing depth, technical replication merging, subsampling, and specific parameters for alignment methods and more information regarding the specific statistical tests and their applicability to the data.This is a promising single-cell study with potentially high significance. The manuscript is well-written, and the analyses are reasonably well-executed. However, the current manuscript is limited by a lack of functional validation and mechanistic insights. The addition of further analyses and experiments, as suggested, will strengthen the conclusions and increase the impact of the work.

We thank the reviewer for their careful reading of the manuscript. As suggested, we have performed additional validation and clarification of the results, as well as a more explicit discussion of their limitations. In addition, we have included a preliminary analysis exploring potential mechanisms that could be coordinating the observed expression patterns of the TcS family (see below). Even though we consider relevant and interesting to experimentally validate these results, given the inherent difficulties in studying multigene families in *T. cruzi*, an organism with a very limited set of molecular biology tools (such as RNAi), further experimental validation of these observations is outside of the scope of this short report.

Regarding the reviewer’s question, we studied if any TcS subgroup could be driving our observations. However, we did not find any correlations indicating that a particular group was associated with any of our findings. We now include TcS group information to Supplementary Table 3.

Regarding technical details, we now included the total number of mapped reads (line 422) and average number of reads mapped per cell (new paragraph in the Methods section, line 432-436).

The technical replicates consist of a single Illumina library that was sequenced in two separate runs. As this approach is expected to be highly reproducible, we merged both runs into a single count table, as stated in line 424. To support this decision, we assessed the concordance between the two sequencing runs and observed an almost perfect correlation between them (Author response image 2).

The subsampling method was originally described in the Figure 2 legend; to better highlight this approach, we have now moved its description to the Methods section (line 456).

The specific kallisto parameters used are stated in Methods (line 418-419). We now included that default options were used unless otherwise specified (line 419-420).

In response to the reviewer’s and editorial team’s request for additional mechanistic insight into the regulatory processes that may be involved in the observed patterns, we have expanded the revised manuscript to discuss how the genomic context of TcS loci could contribute to the observed heterogeneity in TcS expression. As noted in the original version of the manuscript, TcS genes and other surface-protein gene families are largely partitioned into discrete genomic compartments, whose expression has been reported to be regulated by epigenetic control of chromatin-folding domains (doi.org/10.1038/s41564-023-01483-y). However, we previously showed that TcS genes detected in a high proportion of cells are, in most cases, dispersed throughout the genome, arguing against a model in which their preferential expression results from colocalization within a small number of ubiquitously activated chromatin domains. In response to the reviewer’s suggestion, we performed a more detailed analysis of the genomic locations of these TcS genes. We found that many of them are localized within the core compartment (new Figure 5). Because the core compartment is enriched for conserved, housekeeping genes that typically display more constitutive expression (doi.org/10.1038/s41564-023-01483-y), whereas the disruptive compartment is enriched for lineage-specific multigene families associated with variable, stage-specific, and recently reported stochastic expression (doi.org/10.1038/s41467-025-64900-2), our results are consistent with a model in which compartment-specific regulatory mechanisms (in addition to post-transcriptional regulation) influence the differential cellular expression of core- versus disruptive-located TcS genes. We have incorporated these results and discussion in line 301-313 of the revised manuscript.

**Reviewer #3 (Recommendations for the authors):**
(1) The authors consistently refer to gene "expression" but somewhere they should acknowledge that in trypanosomes RNA abundance is less predictive of protein than in most other organisms.

We thank the reviewer for this important comment, highlighting a central challenge when studying trypanosomatid biology. We acknowledge that in most eukaryotes and particularly in *T. cruzi*, where there is a predominant role of post-transcriptional regulation, mRNA levels are not always directly correlated with protein abundance, as previously reported by us and others (10.1186/s12864-015-1563-8, 10.1128/msphere.00366-21, 10.1590/S0074-02762011000300002, 10.1042/bse0510031). Nevertheless, steady-state transcript levels obtained by RNA-seq remain informative for assessing differential gene expression, and this approach has been widely used as a proxy for the study of gene expression profiles in *T. cruzi* (10.7717/peerj.3017, 10.1371/journal.ppat.1005511, 10.1016/j.jbc.2023.104623, 10.3389/fcimb.2023.1138456, 10.1186/s13071-023-05775-4).

It's also interesting to note that recent proteomic analyses (10.1038/s41467-025-64900-2) have revealed substantial heterogeneity in the expression of surface proteins, including trans-sialidases, supporting the idea that the transcriptional heterogeneity we observe reflects a genuine biological feature that propagates to the protein level.

We have now added a sentence to the discussion acknowledging this limitation and discussed the results from Cruz-Saavedra, et al. in linea 266-271 of the revised manuscript.

(2) Line 29, in the abstract there is a strong statement that *T. cruzi* "does not employ antigenic variation". I don't think there is much evidence either way if we are thinking about antigenic variation in the broad sense rather than the extreme model of *T. brucei* VSG switching. Later in the abstract they state that "no recurrent combinations of TcS genes were observed between individual cells in the population", which sounds very much like a form of antigenic variation.

We agree with the reviewer. Indeed, we meant to state that *T. cruzi* does not employ an antigenic variation mechanism such as the one from *T. brucei*. We change this statement as suggested in lines 28 - 32.

(3) Line 29, "relies on a diverse array of cell-surface-associated proteins encoded by large multi-copy gene families (multigene families) essential for infectivity and immune evasion" and lines 55-58 "T. cruzi infection relies on a heterogeneous set of membrane proteins, encoded mainly by large multigene families ... most of which are involved in infection, tropism, and immune evasion". It would be worth adding a bit more detail on the nature and strength of the evidence that Tc "relies on" these various genes or that they are "essential" for infectivity, tropism, and immune evasion.

Because the journal’s short format imposes word limits, we strengthened the original statement by adding specific references that document genomic, transcriptomic and functional evidence linking the major multigene families to infectivity, tropism and immune evasion (doi.org/10.1371/journal.pone.0025914; doi.org/10.1038/nrmicro1351; doi.org/10.1128/iai.05329-11; doi.org/10.1093/nar/gkp172, doi.org/10.1371/journal.ppat.1006767), in line 77.

(4) Line 89, 1088 genes detected per cell - what is this as a % of genes in the genome?

We detected a mean of 1088 genes per cell. Based on the 15,319 annotated protein-coding genes in the reference genome, this represents 7.1% of the *T. cruzi* protein-coding gene complement detected in each cell.

Across the entire dataset, a total of 14,321 genes were detected in at least one cell, representing 93.5% of all annotated protein-coding genes. This suggests that our experiment captured a broad representation of the parasite's transcriptome.

This per-cell detection rate is characteristic of droplet-based scRNA-seq and is consistent with other trypanosomatid studies. For example, the *T. brucei* single-cell atlas (Hutchinson et al., 2021) reported a median detection of 1052 genes per cell. In the case of *T. cruzi*, the recently published pre-print of the *T. cruzi* single cell atlas from Laidlaw & García-Sánchez et al. reported a mean between 298 and 928 genes detected per cell (depending on the sample).

This information is now included in Methods (line 435).

(5) Line 93-94, how many cells were assigned to clusters 0 and 1?

Cluster 0 had 2201 cells and cluster 1 had 824 cells assigned. We have now included these specific numbers in new version of the manuscript (line 114).

(6) Line 96, cluster 2 ama-trypo transitioning parasites - were these observable by microscopy?

We did not perform microscopy specifically to observe or quantify the putative ama/trypo transitioning subpopulation: microscopy was only used as a pre-experiment quality check to verify cell morphology and viability. The inference that cluster 2 reflects ama/trypo transitioning parasites is drawn from the transcriptomic profile (particularly from the pattern of stage-associated marker expression observed in that cluster) and should be considered a hypothesis generated by the data, that merits further analysis, as stated in the manuscript.

(7) Line 106-107, "As expected, single-copy gene expression is high in both amastigotes and trypomastigotes and similar on average between both cell types".(8) Why as expected? For a broad journal it would be useful to explain this. Amastigotes are replicative and trypomastigotes are not, so would we not expect to see some differences that reflect this?(9) What do you mean by the expression being "high"? High compared to what?(10) "Similar on average between both cell types". This does not seem concordant with Figure 1a showing a highly significant difference between ama and trypo.

We thank the reviewer for this helpful request for clarification for broader readers and the observations regarding global expression of single copy and multigene family genes.

Figure 2a is intended as an experimental control where we show that our 10X Genomics data shows the previously reported upregulation of surface protein genes in trypomastigotes. We have now modified the text in order to highlight this (line 129). In turn, Supplementary Figure 1a is shown as a control that this upregulation is not a general feature of trypomastigote cells.

Regarding comment 9, what we meant is that single-copy genes display relatively high expression in both amastigotes and trypomastigotes compared with surface protein-coding genes (see expression values in Figures 2a and Supplementary Figure 1a).

Finally, differential expression between amastigotes and trypomastigotes at the transcriptomic level has been previously studied and has shown that most single copy genes do not show variation, explaining the overall pattern of Supplementary Figure 1a where average expression is similar between stages (mean fold change = 1.1). This is likely due to the fact that these genes are related to basic cellular functions. Genes related to stage specific functions such as replication in amastigotes or normalization effects may be causing the slight, but statistically significant increase observed in overall expression in amastigotes. This contrasts with the pattern observed for multigene families where there is a clear overexpression in trypomastigotes (mean fold change = 1.5).

As observations commented on questions 9 and 10 have been described in previous studies and are not novel nor key points in our results, we decided not to focus on them and modified the text accordingly in lines 129-135.

(11) Line 110, "with high variation". What does "high variation" mean here? Compared to what? For the two metrics (n cells +ve for each gene and total expression level) can they give an average and the SD? It would be useful to know how many parasites the "average" surface (and core) gene is expressed in, or more precisely for which the RNA is above the limit of detection.

We refer to the comparison with the expression profile observed for single-copy genes. This point has now been clarified in the text, and we have included the mean and standard deviation for both TcS multigene family genes and single-copy genes in trypomastigotes for both metrics in the Figure 2 legend. The average and distribution of the number of cells in which each gene is detected are shown in Figure 2c and Supplementary Figure 1a. We also added a reference to this panel at the point in the text where the phenomenon is first described.

(12) Line 134, Figure 2b legend needs more detail - what are num_multigene and z_multigene?

Please see our response to Reviewer 1, Question 6. We have now added a clarification to the legends of Figure 1 and Supplementary Figure 1.

(13) Figure 2c, correct the y-axis legend because it implies your values are log10 transformed. Also, it would be useful to have more markers on the y axis so the reader can better estimate the data ranges.

We thank the reviewer for this observation. We have now corrected the y-axis label and markers.

(14) If the y-axis of Figure 2D started at 0 instead of 0.8 and if Lorenz curves were provided then the reader would probably get a fuller sense of the expression heterogeneity in the dataset. The legend states the differences are statistically significant but the actual p-values are not shown.(15) Line 142-3, more precision is needed on the p-values.

We thank the reviewer for this helpful suggestion. We agree that Lorenz curves provide a clearer representation of expression heterogeneity than the previous plot. Accordingly, we have replaced the original panel (Figure 2d) with Lorenz curves for the groups under comparison, and have made the same change in Supplementary Figure 1d. In addition, we have included gini index values and p-values for all comparisons in Supplementary Table 2.

(16) Figure 3, as in Figure 1a it would be useful to add another UMAP plot to show the two trypo subpopulations.

We thank the reviewer for this suggestion. We have now updated Figure 3 to include a UMAP plot showing the two trypomastigote subpopulations.

(17) What is the observed proportion of broad vs slender trypomastigote morphologies for Dm28c? To be consistent with the speculation at line 162 then wouldn't it need to be approximately 50-50?

The proportions of each trypomastigote subpopulation in the DM28c strain are currently unknown. The only available relevant data come from Brener, 1965 (doi.org/10.1080/00034983.1965.11686277), in which this strain was not included. In the strains analyzed in that study, the relative proportions of broad and slender trypomastigote morphologies were highly variable: across seven strains, broad forms ranged from 18.0% to 77.3%, while slender forms ranged from 2.3% to 71.6%. Given this wide variability and the lack of DM28c-specific data, we cannot assume any expected proportion for this strain.

(18) Line 170, please state how many genes are in the TcS subgroup mentioned here. This is an interesting finding - does this include mostly catalytically active trans-sialidase genes or is it a mixture from across all the subfamilies?

The TcS subgroup with a high frequency of detection comprises 31 genes, none of which belong to the catalytically active Group I trans-sialidases. Instead, this subgroup includes members of Groups II, III, IV, V, VI, and VIII. This information has been added to Supplementary Table 3 and is now stated in the revised manuscript (lines 227 - 228).

(19) Line 175-176, "Gene dropouts might favor random patterns of gene family's detection in scRNA-seq experiments, particularly affecting genes with low expression" - I'm not sure if the authors mean the detection of a gene (or not) in an individual parasite is truly random (pure luck) or whether the term stochastic would be more appropriate because they seem to be referring to randomness around a certain threshold of RNA abundance/stability? They go on to rule this out, at least for TcS genes, essentially arguing that they have something resembling an ON or OFF pattern rather than a spectrum of expression levels. This is potentially very important and could advance the field in a major way, but the fact that so many core and ribosomal genes, which 'should' be always ON, cannot be detected in most cells is a concern. A version of Figure 4B for core and ribosomal genes could be informative - do they show a different pattern to TcS?

Our results reveal a small subset of TcS genes that are frequently detected across cells, a pattern that is not compatible with random detection unless these genes were highly expressed and preferentially captured by random sampling. However, as shown in Figure 4b, many genes expressed at comparable levels are not detected at high frequencies. In line with this, Figure 4c shows that within individual cells, the detected TcS genes exhibit similar expression levels. Finally, we confirmed that this frequently detected subset shows high read counts at the bulk RNA-seq level (Supplementary Figure 2), consistent with the fact that these TcS are frequent in the population even when they are not specially highly expressed within each cell. Taken together, these findings argue against a purely random sampling of TcS genes and support the interpretation that this pattern reflects an underlying biological feature. We agree that further validation will be required. Accordingly, since the initial submission, we have been careful to frame our conclusions conservatively, explicitly noting that dropout remains a limitation of these data that could influence the observed patterns. In the revised version, we have strengthened this point by including a specific statement in the final remarks. Our interpretation is presented as a working hypothesis that is fully compatible with the observations reported here and may be informative for the field. To better reflect this reasoning, we have revised Figure 4b, expanded the discussion, and explicitly included this limitation in the final remarks of the revised manuscript.

(20) Line 238-9, Add details of removing extracellular epimastigotes after cell infections.

Only cellular trypomastigotes collected from the supernatant on day 6 were used for the secondary infection, at a 10:1 parasite-to-cell ratio. After 24 hours, the cultures were washed twice with PBS to remove any remaining extracellular parasites. Under these conditions, i.e. using exclusively trypomastigotes, at this infection ratio, and maintaining the cultures in mammalian medium, we do not expect the presence or survival of extracellular epimastigotes. We have included a sentence in the Methods section clarifying this information in the revised version of the manuscript, line 382.

(21) Line 260, was methanol used to directly resuspend the parasite pellet, or was it resuspended first e.g. in a small volume of PBS?

As described in lines 250-257 of the original manuscript, parasites were washed and resuspended in DPBS before methanol fixation. Methanol fixation was then carried out according to the 10X Genomics Methanol Fixation Protocol. We have now emphasized this more clearly in the revised text in line 400.

(22) What was the doublet rate?

We identified and removed 41 doublets, all belonging to cluster 2, and retained 3,151 singlets for downstream analysis (total cells before removal = 3,192). The resulting doublet rate was 1.28%. We have included a sentence in the Methods section clarifying this information in the revised version of the manuscript, line 439 -440.

(23) What was the frequency of rRNA and kDNA-derived reads?

Approximately 4.02% of the reads were derived from kDNA sequences, while 1.10% corresponded to rRNA-derived reads (Author response image 4).

**Author response image 4. sa4fig4:** Percentage of mitochondrial and ribosomal rRNA derived reads.